# The planarian dorsal–ventral boundary regulates anterior–posterior axis growth and patterning

Chloe L. Maybrun[1,2], Isaac M. Oderberg[1,2], Michael A. Gaviño[1,2], Thomas F. Cooke[1,3], Kyungyong Choi[4], Jongyoon Han[4,5], Peter W. Reddien [1,2,3]*

1 Whitehead Institute for Biomedical Research, Cambridge, Massachusetts, United States of America, 2 Department of Biology, Massachusetts Institute of Technology, Cambridge, Massachusetts, United States of America, 3 Howard Hughes Medical Institute, Massachusetts Institute of Technology, Cambridge, Massachusetts, United States of America, 4 Research Laboratory of Electronics and Department of Electrical Engineering and Computer Science, Massachusetts Institute of Technology, Cambridge, Massachusetts, United States of America, 5 Department of Biological Engineering, Massachusetts Institute of Technology, Cambridge, Massachusetts, United States of America

* reddien@wi.mit.edu

## Abstract

Regeneration can involve the coordination of pattern formation in an outgrowth with the spatial pattern of pre-existing tissues, such as along body axes. Planarian adult axis patterning serves as a robust context for uncovering the mechanisms of such pattern integration. We investigated how the dorsal–ventral boundary (DVB), which surrounds the animal periphery at the dorsal–ventral (DV) median plane, regulates anterior–posterior (AP) axis growth and patterning. We define a spatial DVB gene expression atlas that includes genes encoding signaling, adhesion, and transcription factors. Wnt inhibition results in anterior positional information induction and ectopic head formation that is restricted to the DVB. DVB can be transplanted, and DVB identity can be experimentally induced at ectopic locations. Ectopic DVB is competent for anterior positional identity induction following Wnt inhibition, enabling the generation of animals with ectopic heads at experimentally dictated locations. DVB removal blocks the anteriorization that normally follows Wnt inhibition and prevents anterior positional information expression during head regeneration. Anterior positional information induction at the DVB after Wnt inhibition occurs independently from anterior pole formation, which promotes head patterning in regeneration. Our findings reveal a hierarchical model of pattern integration across body axes in which DV patterning is central by producing a DVB with competence to direct formation of large AP axis regions. This mechanism enables coordination of orthogonal positional information in the context of regeneration.

## Introduction

Many organisms can regenerate appendages or significant parts of body axes [1,2]. During regeneration, tissues must be formed in the correct position within the body and properly oriented relative to remaining tissues. This attribute of regeneration poses a unique patterning challenge in biology. How positional information is integrated across the axes of new and pre-existing, adult tissues is therefore a central and understudied question in the field of regeneration.

---

---

**Data availability statement:** The scRNA-seq data generated in this study have been deposited in the NCBI Sequence Read Archive (SRA) under the accession number PRJNA1299474.

**Funding:** This work was supported by the Eleanor Schwartz Charitable Foundation (to PWR), the National Institutes of Health (R35 GM145345 to PWR), and the Howard Hughes Medical Institute (PWR). The funders had no role in study design, data collection and analysis, decision to publish, or preparation of the manuscript.

**Competing interests:** The authors have declared that no competing interests exist.

**Abbreviations:** AFW, anterior-facing wound; AP, anterior–posterior; BSA, bovine serum albumin; dsRNA, double-stranded RNA; DV, dorsal–ventral; DVB, dorsal–ventral boundary; PCGs, position control genes; scRNA-seq, single-cell RNA sequencing; UMAP, uniform manifold approximation and projection.

Planarians can regenerate from a large array of injuries, including from small fragments of the body [3]. The planarian body plan is organized along three axes: anterior–posterior (AP), dorsal–ventral (DV), and medial–lateral (ML). Positional information refers to factors that influence the regional identity and organization of cell types, organs, and appendages [4]. Positional information across body axes, such as during animal development, promotes organization of the body plan [5–8]. Positional information in adult planarians includes the products of genes referred to as position control genes (PCGs). PCGs are constitutively and regionally expressed along one or more body axes and either have an abnormal patterning phenotype when inhibited or are predicted to be part of planarian patterning pathways by homology [9]. Most PCGs show constitutive regional expression predominantly in planarian muscle [9].

In development, most animals utilize canonical Wnt signaling to control pattern formation on the AP axis, whereas Bmp signaling is used to pattern the DV axis [5–7]. The planarian AP axis is patterned largely by canonical Wnt signaling [10]. A system of posteriorly expressed Wnt ligands and anteriorly expressed Wnt inhibitors establishes a posterior-to-anterior gradient of β-catenin-1 signaling activity that is required for the establishment and maintenance of regional tissue identity [11–20]. RNAi of *β-catenin-1* causes heads to regenerate in place of tails after transverse amputation and induces the formation of ectopic heads in uninjured animals [11–13]. Two FGFRL-Wnt modules and Src regulate planarian head and trunk patterning [21–23]. Bmp signaling controls planarian DV patterning [10]. The expression of the Bmp ligand-encoding *bmp4* gene is restricted to dorsal muscle in a medial-to-lateral gradient [9,24–26]. RNAi of *bmp4* or the downstream pathway component *smad1* results in progressive ventralization in uninjured animals [25–27]. During normal regeneration, major regions of body axes are generated with the correct orientation relative to each other in the blastema (regenerative outgrowth) and to pre-existing patterns within the amputated fragment, suggesting the existence of mechanisms to coordinate axial regeneration.

*bmp4* is required for regeneration of pattern along the AP axis [28]. The possibility that DV pattern informs the location of AP axis establishment was proposed in theoretical models of planarian patterning [29,30] and is supported by the following observations. The anterior pole, a cluster of specialized muscle cells that organizes pattern during head regeneration [31–33], forms at the DV median plane [34]. This region of the animal, sometimes called the dorsal–ventral boundary (DVB), is defined by a zone of epidermal and sub-epidermal gene expression at the lateral edge of the animal, where the dorsal and ventral sides meet. This DVB location is set by a patterning process on the DV axis involving dorsal Bmp signaling [25–27]. We also note that ectopic head formation after *β-catenin-1* RNAi occurs only at the animal lateral edge (body margin) [11–13]. DV confrontation can promote axis formation in certain patterning contexts in other organisms. In *Drosophila*, for instance, interactions between dorsal and ventral cells in the wing imaginal disc drive wing growth [35,36]. Genetic manipulations that create an ectopic boundary between dorsal and ventral cells in *Drosophila* can lead to formation of an ectopic outgrowth of the wing blade. DV contact during wound closure has been proposed to promote planarian blastema

formation [37,38]. Whether diverse organisms across the animal kingdom such as those capable of whole-body regeneration have a DVB present in the adult stage that regulates regeneration, and the mechanistic roles of any such adult DVBs in regenerative contexts remain poorly addressed. The presence of a region at the planarian DV median plane with a candidate regulatory role in regeneration therefore makes planarians an attractive venue for studying the regenerative role and nature of an adult DVB.

In this work, we find that the DVB is sufficient and necessary for planarian head formation in a low Wnt environment. By manipulating the pattern of the DVB, it is possible to induce heads to grow out of the dorsal side of the animal or to block head positional information expression from either the lateral animal edge in a Wnt-low environment or at anterior-facing wounds (AFWs). Our findings support a hierarchical model of pattern integration across body axes in regeneration, with the lateral DV median plane orchestrating growth and patterning of large AP axis regions.

## Results

### The AP and DV axes maintain pattern largely independently in homeostasis

The maintenance of adult positional information in planarians is important for tissue turnover and for regeneration after injury [10]. To maintain positional information with suitable spatial relationships between axes during homeostatic tissue maintenance, positional information along one axis could, in principle, require input from another axis. Inhibition of *β-catenin-1* results in homeostatic anteriorization [11–13] (S1A Fig). However, DV asymmetric tissues [12] and both dorsally biased (*bmp4*, *nlg-8*) and ventrally biased (*admp, nlg-7*) PCG expression remained unchanged after *β-catenin-1* RNAi (Figs 1A, S1B, and S1C), suggesting that Wnt signaling is not overtly required for the maintenance of DV pattern during adult tissue turnover. *bmp4*, the inhibition of which leads to homeostatic ventralization [25–27] (S1D Fig), has been proposed to integrate patterning of the DV and AP axes by suppressing Wnt [39]. In agreement with findings in Clark (2023) [39], we found that the domain of *wnt1+* cells (the posterior pole) was expanded after 45 days of *bmp4* RNAi (Figs 1B and S1E). The *fz4-1* expression domain was similarly lengthened in *bmp4* RNAi animals. However, no overt changes in the *wntP-2* or *ndl-3* expression domains were apparent (Fig 1B). The anterior pole, labeled by *notum*, was expanded, albeit to a lesser extent than the posterior pole (Figs 1B, S1F, and S1G). However, no dramatic changes in anterior PCG expression domains (*sFRP-1*, *ndl-5*) were observed (Figs 1B and S1F), as previously reported [39]. Furthermore, the relative order of PCG expression domains along the AP axis remained largely unchanged after *bmp4* RNAi, suggesting that Bmp signaling is not strictly required for the maintenance of broad AP pattern, but might limit the extent of the posterior-most patterning domain of the AP axis. These results point towards the possibility that axis integration might predominantly be accomplished during regeneration or during ectopic head formation in a low Wnt environment, with pattern thereafter largely stably maintained by each axis independently during cell turnover.

### Organization of PCG expression at the DVB in regeneration

During head regeneration, anterior pole cells labeled by *notum* transcripts form a focus at the DVB [34]. We used epidermal cells expressing *laminB*, which mark the lateral DV median plane [38,40], as a reference point for the DVB. At 96 hours post-amputation (hpa), this DVB location was positioned ventral to the dorsally high *bmp4* domain in muscle and was coincident with the known location of *admp* expression in muscle [9,41,42] at the lateral DV median plane (Figs 1C and S1H). Expression of the anterior PCGs *sFRP-1*, *ndl-4*, and *ndl-5* was centered around the DVB (Figs 1C and S1I) and showed peaked behavior on the dorsal and ventral sides that juxtapose the *laminB+* DVB location (Figs 1D and S1J). The mature posterior pole labeled by *wnt1* is dorsally biased [13,16,43] but formed a cluster near the DVB at 96 hpa during tail regeneration (Figs 1C and S1I). Expression of the posterior PCGs *wnt11-1* and *fz4-1* was centered around the *laminB+* DVB (Figs 1C and S1I) and showed peaked behavior on the dorsal and ventral sides juxtaposing the DVB (Figs 1D and S1K), similar to the case of anterior PCGs. These results connect patterning processes involving PCG expression at the

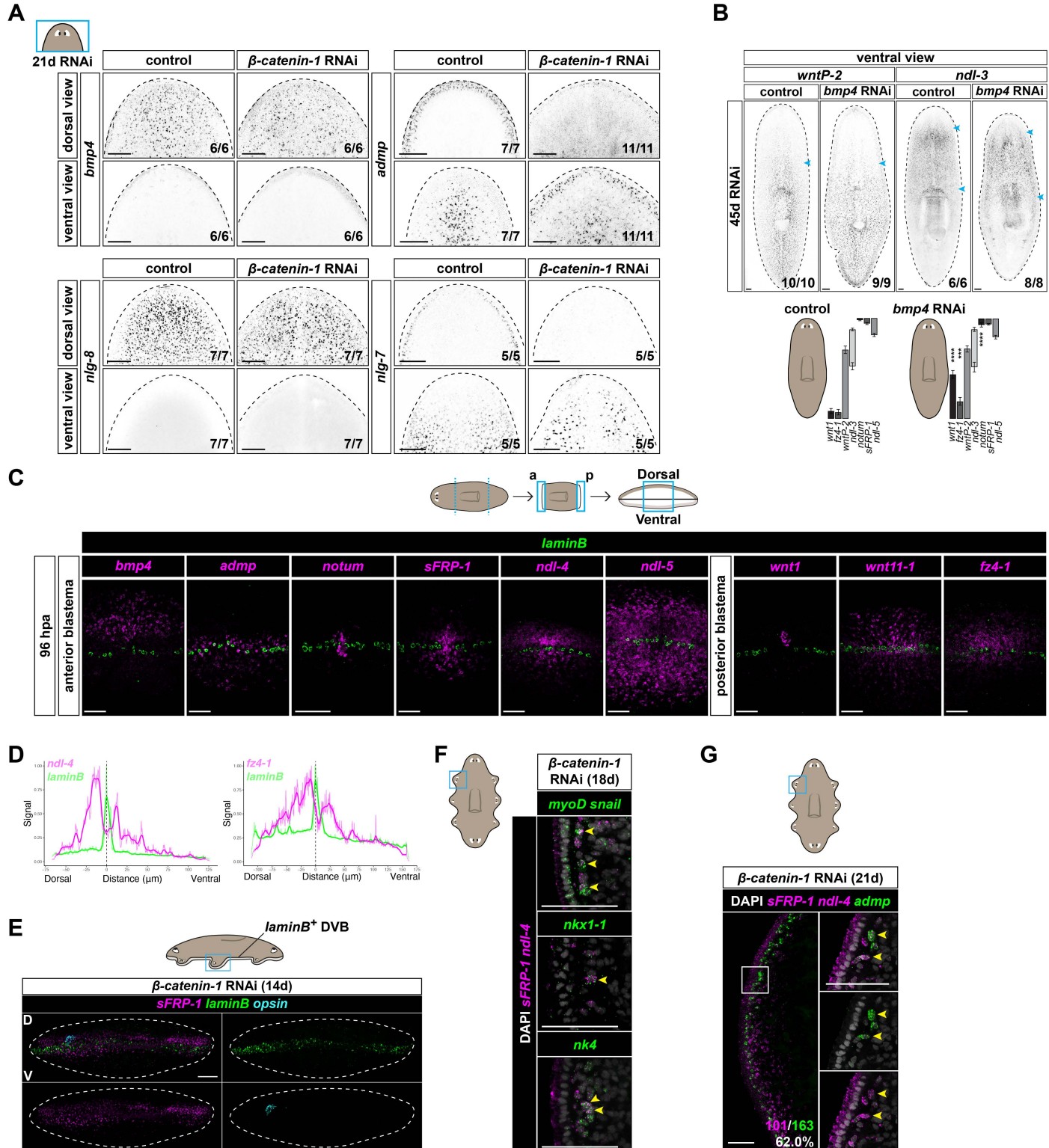

**Fig 1. Ectopic heads form at the DVB after β-catenin-1 RNAi. A)** Dorsally biased (*bmp4*, *nlg-8*) and ventrally biased (*admp*, *nlg-7*) PCG expression after *β-catenin-1* RNAi (21d). **B)** PCG expression domains in *bmp4* RNAi animals (45d). Blue arrow denotes anterior or posterior boundary of PCG expression domain. Diagrams show the mean PCG domain length ± standard deviation relative to body length in *bmp4* RNAi animals. Six to seventeen

animals per PCG were measured. Data points from individual animals are listed in S1 Data. Significant domain shifts are marked: *$p < 0.05$, **$p < 0.01$, ***$p < 0.001$, ****$p < 0.0001$ by two-tailed t-tests. **C)** The epidermal *laminB*+ DVB is positioned ventral to the dorsally high *bmp4* expression domain in muscle and is coincident with the known location of lateral *admp* expression in muscle. Anterior and posterior PCG expression is concentrated near the DVB during regeneration (96 hpa). **D)** Fluorescence intensity along a line is plotted and smoothed data is represented with a thick stroke. The foci of anterior (*ndl-4*) and posterior (*fz4-1*) PCG expression show peaked behavior on the dorsal and ventral sides juxtaposing the *laminB*+ DVB. Individual data points are listed in S1 Data. **E)** Ectopic anterior PCG (*sFRP-1*) expression near the DVB after *β-catenin-1* RNAi (14d). D, dorsal; V, ventral. **F)** Ectopic anterior PCG (*sFRP-1*, *ndl-4*) expression after *β-catenin-1* RNAi (18d) is detected in longitudinal (*myoD*+), circular (*nkx1-1*+), and lateral DV (*nk4*+) muscle at the lateral animal edge. **G)** Co-expression of anterior PCGs (*sFRP-1*, *ndl-4*) and the patterning molecule *admp* in muscle at the DVB after *β-catenin-1* RNAi (21d). Number of *admp*+ cells that co-localize with *sFRP-1* and *ndl-4* transcripts (magenta) out of total *admp*+ cells (green) is shown. *n* = 4 animals. Colored boxes, area depicted in photos. Scale bars, 100 μm (**A**, **B**, **E**), 50 μm (**C**, **F**, **G**).

head and tail tips to the location of the DVB and support a potential mechanism for axis integration [34] wherein the DVB serves as a critical landmark to guide proper regeneration along the AP axis.

### Ectopic heads form at the DVB after *β-catenin-1* RNAi

Ectopic heads form at the animal periphery after *β-catenin-1* RNAi, with the occasional exception of a head near the planarian pharynx, during a process of homeostatic tissue turnover [11–13]. *β-catenin-1* RNAi resulted in ectopic focal expression of the anterior PCGs *sFRP-1* and *ndl-4* laterally, centered at the *laminB*+ DVB (Figs 1E and S1L). Anterior PCGs showed peaked ectopic expression on the dorsal and ventral sides juxtaposing the DVB in some regions (S1L Fig). Ectopic anterior PCG (*sFRP-1*, *ndl-4*) expression occurred in multiple muscle subtypes present near the DVB [44,45], including in longitudinal muscle (muscle that is oriented along the AP axis), circular muscle (muscle that is oriented along the ML axis), and lateral DV muscle (muscle that is oriented along the DV axis at the body margin) (Figs 1F and S1M). Ectopic anterior PCG (*sFRP-1*, *ndl-4*) expression in *β-catenin-1* RNAi animals occurred in a large fraction (62.0%) of muscle cells expressing the DVB PCG *admp* [9,41,42] (Fig 1G). Ectopic anterior PCGs (*sFRP-1*, *ndl-4*) were co-expressed with additional muscle-enriched genes at the DVB, including *lactadherin* (dd_5463, identified below) and *netrin-1* [46,47] (S1N Fig). Although anterior PCG (*sFRP-1*, *ndl-4*) expression was concentrated in foci at the DVB after *β-catenin-1* RNAi, sporadic cells expressing anterior PCGs were also detected dispersed (not in foci) in the animal (S1A Fig). Hereafter, we analyze specifically the anterior PCG expression that occurred in a concentrated manner at the DVB in *β-catenin-1* RNAi animals.

### A spatial atlas of gene expression, including genes encoding cell surface and signaling molecules, in muscle at the DVB

Given the potential significance of the planarian DVB in patterning, and the fact that its molecular determinants are poorly understood, we sought to define a spatial atlas of gene expression for muscle at the DVB. We performed single-cell RNA sequencing (scRNA-seq) of the lateral edges of animals to enrich for DVB cells (S2A and S2B Fig). All major muscle subtypes at the DVB were represented in our data (S2C Fig). Known markers of the body margin from prior literature (*admp* [41,42], *egf-6* [48], *wnt-5* [14,16], *unc5-B* [46], *ephR-2* [25], *netrin-1* [46,47], *follistatin* [49,50], *noggin-1* [25,41], and *netrin-3* [44,47,51]) displayed expression predominantly in longitudinal muscle at the DVB (S2C Fig). Some DVB markers also displayed expression in lateral DV muscle (*nk4*+) and/or circular muscle (*nkx1-1*+) (S2C Fig), but subclustering data from circular muscle did not separate cells by their DV position of origin (S2D Fig).

Subclustering scRNA-seq data from longitudinal muscle cells identified four total subclusters. Transcripts known to be expressed at the DVB were enriched in subcluster 1 (Figs 2A, S2E, and S2F), suggesting genes with enriched expression in this subcluster could represent DVB-expressed genes. Thirty-two total genes had significantly enriched expression in subcluster 1; nine of these genes were previously characterized as being expressed at the body margin, with 23 new DVB genes. We assessed the expression of these 32 genes by FISH, together with *laminB* as a common spatial reference marker. This generated a spatial atlas of gene expression in muscle at the DVB (Figs 2B–2D, S2G, and S2H). Previously

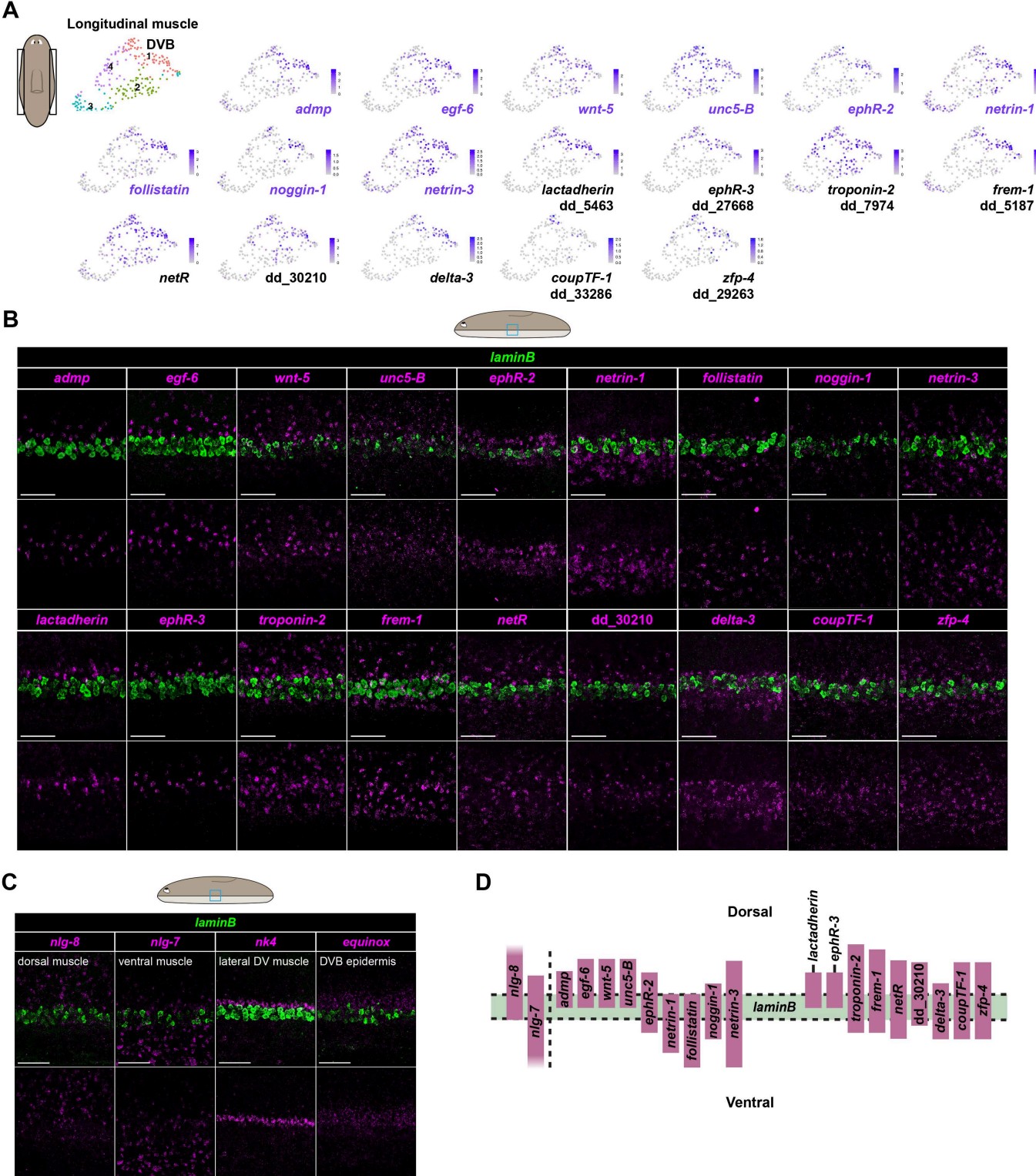

**Fig 2. Spatial DVB gene expression atlas in muscle. A)** Cartoon shows region of the animal isolated for scRNA-seq. Subclustering of longitudinal muscle cells from scRNA-seq data reveals a DVB cell population in UMAP plots. Novel markers of the DVB are labeled in black. **B)** FISH characterization of the DVB region. Images show the expression of genes along the DV axis relative to the spatial domain of the canonical epidermal marker of the

DVB, *laminB*. **C)** FISH images show the expression of genes that label dorsal (*nlg-8*) and ventral (*nlg-7*) muscle, lateral DV muscle (*nk4*), and a DVB epidermis domain (*equinox*) relative to *laminB*. **D)** Cartoon representation of expression domains at the DVB. Dorsal (*nlg-8*) and ventral (*nlg-7*) PCGs are shown as graded columns. Scale bars, 50 µm (**B**, **C**).

identified DVB markers were expressed in either dorsal (*admp*, *egf-6*, *wnt-5, unc5-B, ephR-2*), ventral (*netrin-1, follistatin*), overlapping (*noggin-1*), or both dorsal and ventral (*netrin-3*) domains of the DV axis relative to *laminB* (Fig 2B—2D). In addition to these known DVB markers, nine additional genes (*lactadherin* (dd_5463), *ephR-3* (dd_27668, dd_39545), *troponin-2* (dd_7974), *frem-1* (dd_5187), *netR,* dd_30210, *delta-3*, *coupTF-1* (dd_33286), and *zfp-4* (dd_29263, dd_21349)) were also expressed in particular domains relative to *laminB* at the DVB (Figs 2B—2D, S2I, and S2J). Lactadherin is a secreted protein that can bind anionic phospholipids [52]. Ephrin receptors mediate the effects of Ephrins, a family of membrane-bound signaling ligands that promote a variety of cellular responses including attraction, repulsion, and migration [53]. *frem-1* encodes an extracellular matrix protein that contributes to epithelial-mesenchymal integrity in development [54]. Netrin receptors mediate the attractant and repellent properties of secreted Netrins that guide cell and axon migration in development [55]. dd_30210 encodes a novel protein with no characterized PFAM domains. *delta-3* encodes a membrane-bound Notch signaling ligand. Notch signaling regulates diverse developmental processes, including cell fate determination [56]. COUP transcription factors encode orphan nuclear receptors that regulate diverse cellular processes in development [57]. *zfp-4* encodes a novel zinc finger protein. Using our scRNA-seq data, we calculated the Pearson correlation coefficient for all pairwise comparisons of the 18 genes in the DVB atlas with expression detected by FISH. Within longitudinal muscle cells at the DVB, the expression of many genes showed a high degree of correlation ($r > 0.5$) (S2K Fig), indicating that individual muscle cells co-express multiple markers of the DVB. There were 14 genes with transcripts enriched in longitudinal muscle subcluster 1 (S2E and S2F Fig) that did not display clear DVB-restricted expression by FISH, either because expression was broad or not detected.

Taken together, the molecular data indicate that the DVB, often defined anatomically, can be viewed as an intricately patterned domain of gene expression that in muscle includes a large array of genes encoding signaling factors, cell-surface molecules, and transcription regulators (Figs 2D, S2G, and S2H). DVB muscle is thus a prominent but poorly understood pattern element of the planarian body plan.

### Ectopic DVB is permissive for the formation of an anterior region of the AP axis after *β-catenin-1* RNAi

To determine whether the DVB is sufficient to promote the formation of an anterior region of the AP axis after *β-catenin-1* RNAi, we endeavored to generate animals with ectopic DVB. Transplantation of tissue with reversed DV polarity has been shown to induce ectopic DVB, labeled by epidermal expression of *laminB*, in goblet-shaped outgrowths [38,58]. By contrast, similar tissue transplantation without DV reversal leads to tissue integration without DVB formation. We used this approach to generate outgrowths with ectopic DVB in the post-pharyngeal region of the animal (Figs 3A and S3A). These outgrowths contained DVB muscle gene expression signatures, including for genes encoding the secreted molecules *admp, lactadherin*, and *frem-1*, indicating that both epidermis and underlying muscle acquire an ectopic DVB identity in outgrowths (Figs 3B and S3B). These outgrowths had AP character typical of the location of the transplant [38] (S3C Fig) and lacked anterior identity (Fig 3C). After allowing dorsal-to-ventral (D-V) transplant outgrowths to form, we initiated *β-catenin-1* RNAi. Whereas head structures are normally constrained to the periphery of *β-catenin-1* RNAi animals, D-V transplant outgrowths developed into goblet-shaped head-like structures marked by the presence of eyes in the middle of the animal (Fig 3A). These outgrowths (both dorsal and ventral outgrowths), after *β-catenin-1* RNAi, expressed anterior PCGs (*sFRP-1, ndl-4*) and contained *opsin*⁺ photoreceptor cells (Figs 3C and S3D). By contrast, dorsal-to-dorsal (D-D) transplant animals simply healed the wound ($n = 13/13$) and these transplantation sites did not express anterior PCGs or transform into heads following *β-catenin-1* RNAi (Fig 3A and 3D).

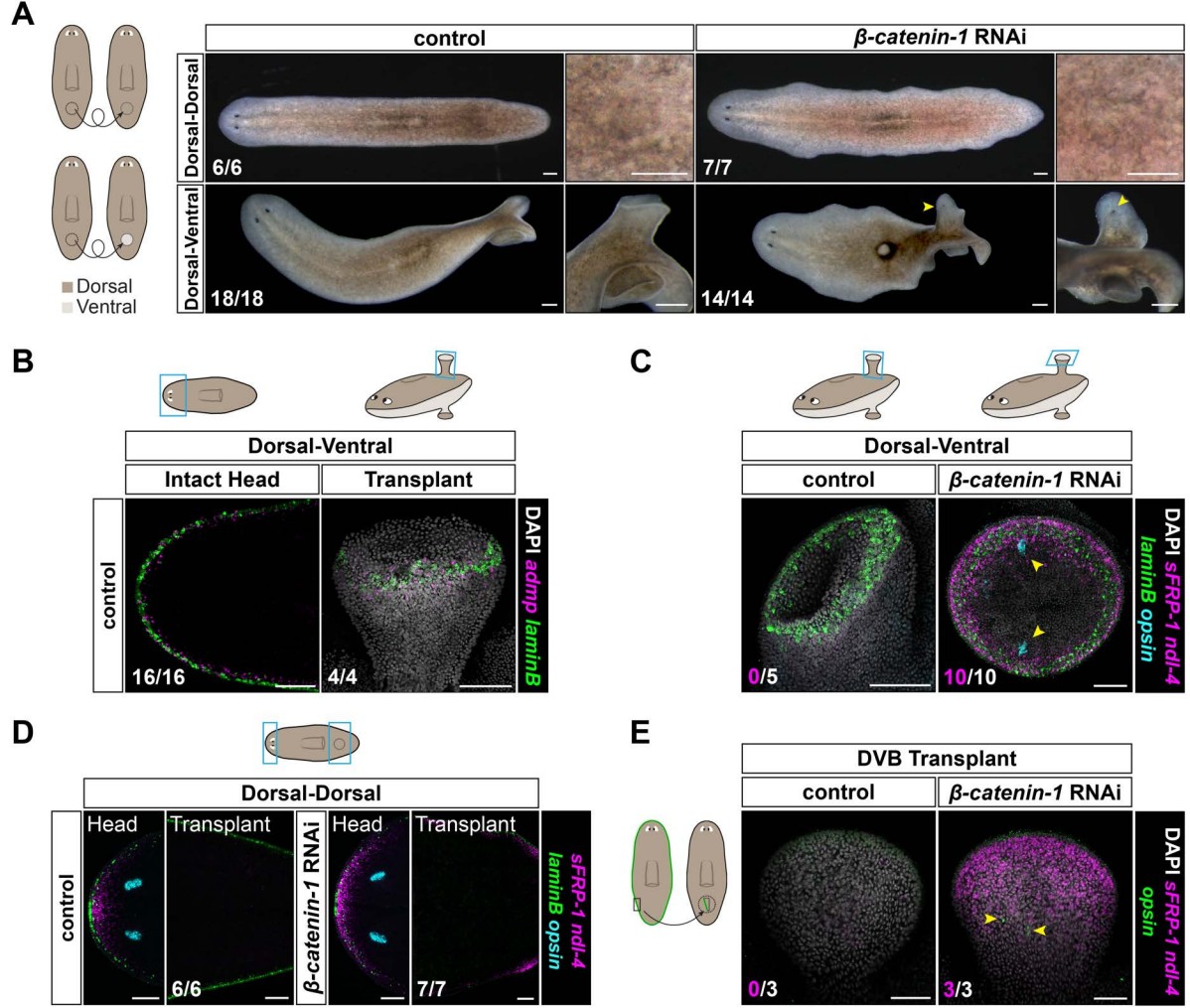

**Fig 3. Ectopic DVB generated by transplantation is competent to induce formation of an anterior region of the AP axis after *β-catenin-1* RNAi.**
**A)** Transplant animals that received tissue grafted with reversed DV polarity develop goblet-shaped outgrowths. After *β-catenin-1* RNAi (10–14d), these outgrowths form head-like structures. Yellow arrowhead, eye. **B)** D-V transplant outgrowths show epidermal (*laminB*) and muscle (*admp*) DVB gene expression. **C)** D-V transplant outgrowths express anterior PCGs (*sFRP-1*, *ndl-4*) and form eyes (*opsin*) after *β-catenin-1* RNAi (10–14d). Yellow arrowhead, eye. **D)** D-D transplants do not result in ectopic DVB (*laminB*) or anterior PCG (*sFRP-1*, *ndl-4*) gene expression after *β-catenin-1* RNAi. **E)** DVB transplant animals develop dorsal outgrowths. After *β-catenin-1* RNAi (10d), these outgrowths express anterior PCGs (*sFRP-1*, *ndl-4*) and form eye cells (*opsin*). Colored boxes, area depicted in photos. Scale bars, 250 μm (**A**), 100 μm (**B, D**), 50 μm (**C, E**).

To complement the use of transplantation to generate ectopic DVB, we next transplanted the DVB region itself. Transplantation of a small posterior DVB-containing segment of the animal margin to the midline of a recipient animal induced the formation of a dorsal outgrowth in the recipient. These outgrowths expressed anterior PCGs (*sFRP-1*, *ndl-4*) and contained *opsin*+ photoreceptor cells after *β-catenin-1* RNAi (Fig 3E), indicating that ectopically positioned DVB is competent to promote anterior patterning after Wnt inhibition.

As an independent method to generate ectopic DVB, we utilized *smad1* RNAi. *smad1* encodes a downstream mediator of Bmp signaling and has a prominent role in planarian DV patterning [25]. Inhibition of *bmp4* or *smad1* leads to homeostatic ventralization [25–27], with the appearance of ectopic *laminB*+ DVB cells on the dorsal surface of animals, in some cases in organized stripes (Fig 4A). After allowing *smad1* RNAi animals to develop ectopic DVB, we inhibited *β-catenin-1*

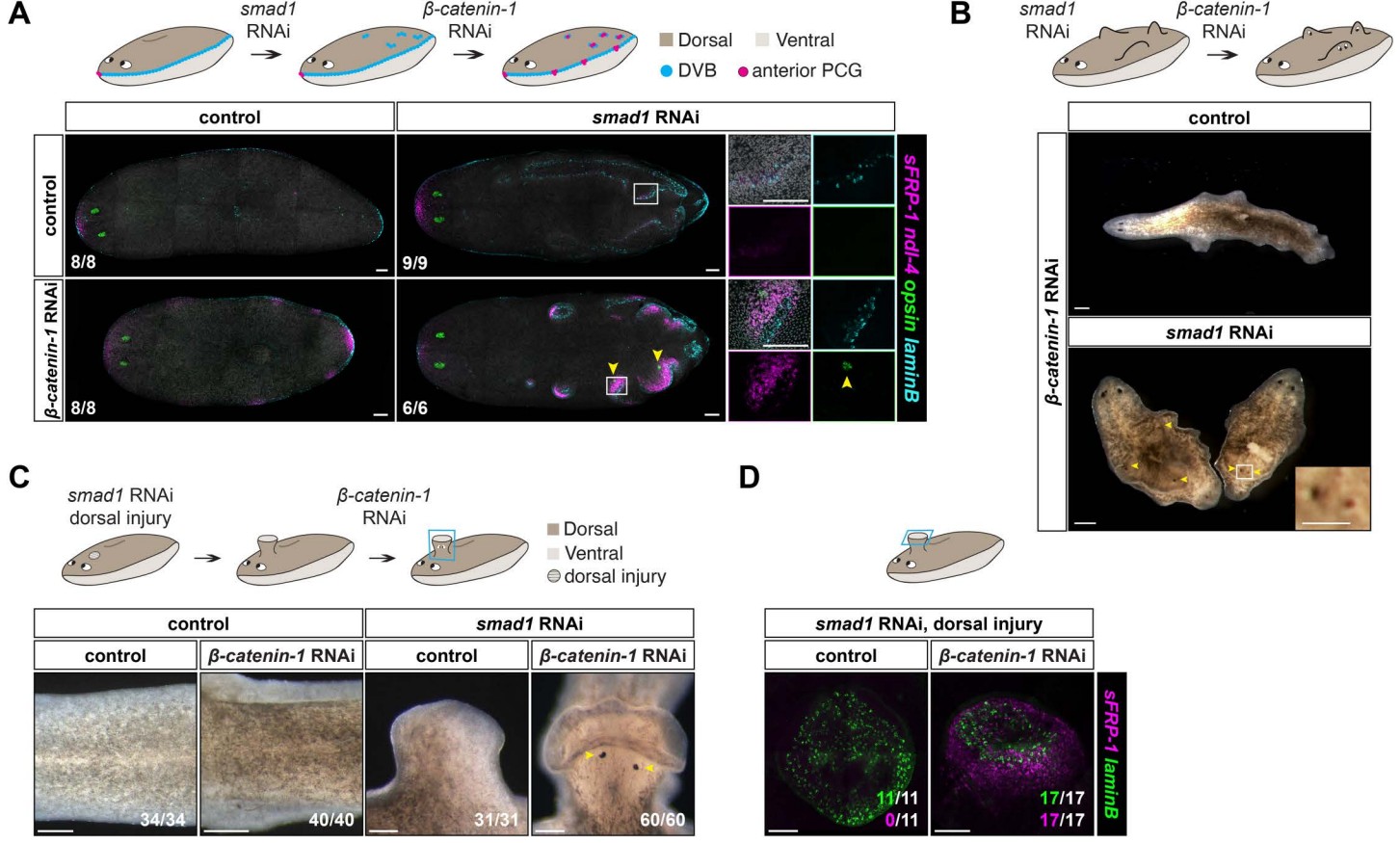

**Fig 4. Ectopic DVB generated by DV axis perturbation is competent to induce formation of an anterior region of the AP axis after β-catenin-1 RNAi. A)** *smad1* RNAi animals develop ectopic DVB (*laminB*) that is anteriorized (*sFRP-1 ndl-4*+) after *β-catenin-1* RNAi (10d). **B)** *smad1* RNAi causes head-like structures to form on the dorsal surface of animals after *β-catenin-1* RNAi (21d). Yellow arrowhead, eye. **C)** *smad1* RNAi animals develop outgrowths at the site of dorsal injury. After *β-catenin-1* RNAi (12d), these outgrowths form head-like structures. Yellow arrowhead, eye. **D)** Outgrowths in injured *smad1* RNAi animals show epidermal DVB (*laminB*) gene expression. After *β-catenin-1* RNAi (12d), these outgrowths express the anterior PCG *sFRP-1*. Colored boxes, area depicted in photos. Scale bars, 100 μm (**A**, **D**), 250 μm (**B**, **C**).

with RNAi. These *smad1; β-catenin-1* double RNAi animals expressed anterior PCGs (*sFRP-1*, *ndl-4*) and developed *opsin*+ photoreceptor cells at sites of ectopic DVB, indicative of head patterning (Figs 4A and S3E). The ectopic anterior PCG expression observed traced the variable and irregular pattern of the ectopic DVB on the dorsal side of these animals (Fig 4A). *smad1; β-catenin-1* double RNAi animals ultimately formed heads that emerged from the dorsal surface of the animal (Fig 4B).

We realized it could be possible to use *smad1* RNAi to induce head formation by design in a desired location. We performed a dorsal injury in the pre-pharyngeal region three days after initiating *smad1* RNAi, triggering outgrowth formation at the injury site that displayed ectopic *laminB*+ DVB gene expression. After subsequent *β-catenin-1* inhibition, the outgrowths were converted to head-like structures (Fig 4C and 4D). Together, these results are consistent with a model in which muscle at the planarian DVB is competent to activate anterior PCGs and support formation of an anterior region of the AP axis in a low Wnt environment. By combining methods to generate ectopic DVB at designated locations with Wnt pathway inhibition, head location and shape can be controlled to yield a large array of body plans.

**The DVB region is required to promote anterior PCG expression in a low Wnt environment**

To test whether the DVB is required to initiate head formation in the context of *β-catenin-1* RNAi, we inflicted either a lateral or internal injury after RNAi (Figs 5A and S4A). Anteriorization after *β-catenin-1* RNAi requires new cell production [18,20], and loss of tissue elicits a localized increase in mitoses at the wound site [59,60]. However,

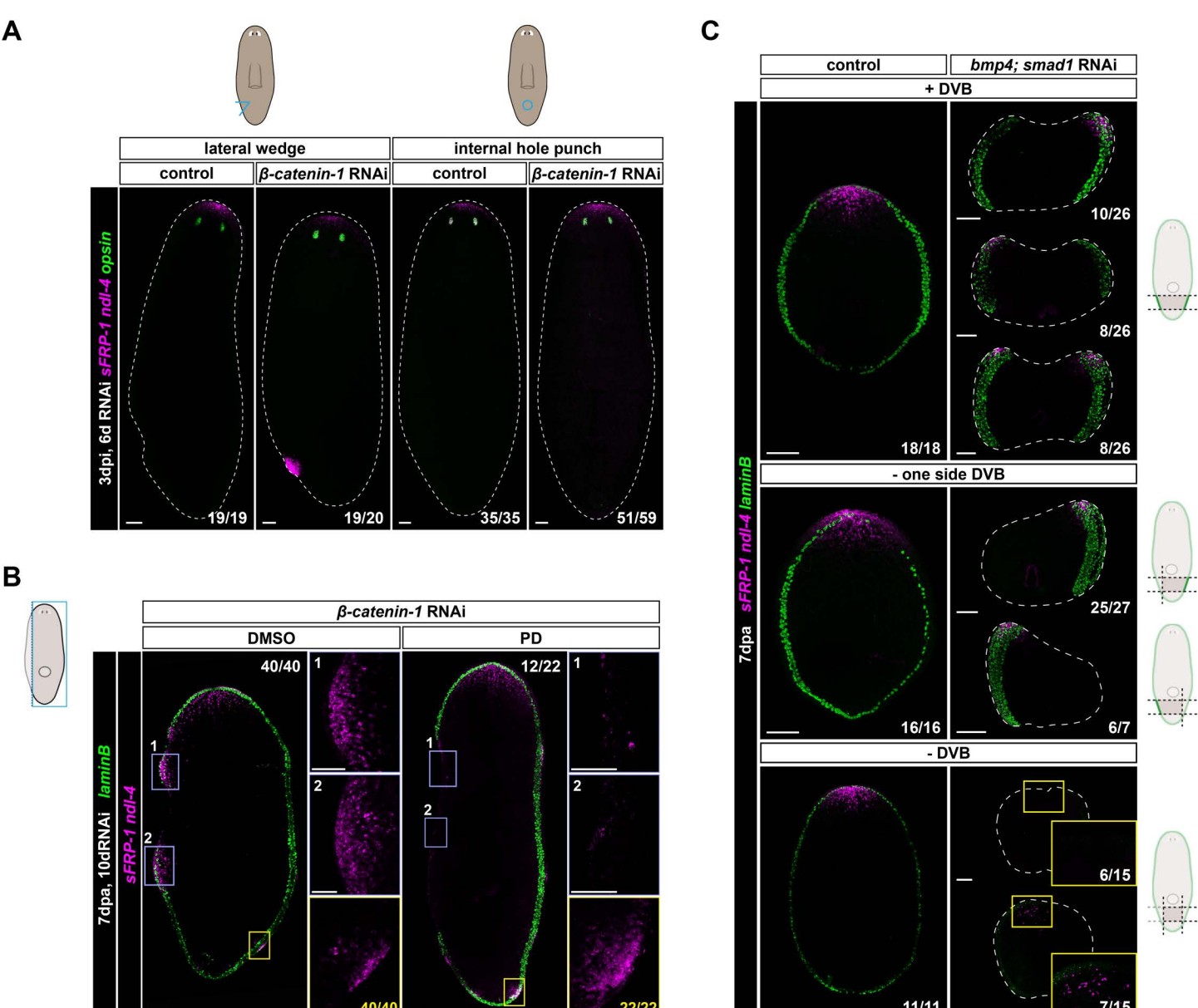

**Fig 5. The DVB region is required to promote anterior PCG expression in a low Wnt environment. A)** *β-catenin-1* RNAi animals after a lateral wedge cut develop a focus of anterior PCG (*sFRP-1*, *ndl-4*) expression at the wound. **B)** Treatment with the MEK inhibitor PD blocks lateral anterior PCG (*sFRP-1*, *ndl-4*) expression after *β-catenin-1* RNAi and DVB removal (*n* = 12/22). Anterior PCGs are expressed in foci at the intact DVB on the uninjured side of PD-treated *β-catenin-1* RNAi animals (*n* = 22/22). **C)** *bmp4; smad1* RNAi post-pharyngeal fragments express anterior PCGs (*sFRP-1*, *ndl-4*) at the anterior-facing wound despite no DVB (*laminB*) regeneration. However, anterior PCG expression is limited to the lateral edge of the fragment near one or two pre-existing, intact DVB domains. *bmp4; smad1* RNAi fragments with both DVB domains removed show little anterior PCG expression. Colored boxes, area depicted in photos. Scale bars, 100 μm (**A–C**), 50 μm (**B**, inset).

only tissue removal at the lateral edge of *β-catenin-1* RNAi animals was sufficient to reliably induce anterior PCG expression at the wound (Fig 5A). An internal hole-punch that removed tissue did not cause anterior PCG expression in most cases ($n = 51/59$) (Figs 5A and S4B), suggesting that signals at the DVB are necessary to robustly initiate anterior patterning.

We next surgically removed the DVB (parasagittal amputation) and blocked its regeneration by inhibiting Erk signaling using PD0325901 (PD) (Figs 5B, S4C, and S4D). Erk signaling is required to promote wound-induced signaling that culminates in a regenerative response [61]. Erk signaling was not required, however, for the homeostatic outgrowth that accompanies ectopic head formation after *β-catenin-1* RNAi (S4E Fig). DMSO-treated *β-catenin-1* RNAi animals regenerated the DVB, labeled by *laminB*, and expressed anterior PCGs (*sFRP-1*, *ndl-4*) in large foci on the side that was regenerating, whereas PD-treated *β-catenin-1* RNAi animals did not regenerate the DVB and little anterior PCG expression was detected on this side in most cases ($n = 12/22$) (Fig 5B). In some cases ($n = 9/22$), anterior PCG (*sFRP-1, ndl-4*) expression was detected on the injured side of PD-treated *β-catenin-1* RNAi animals, despite no regeneration of epidermal DVB (*laminB*) (S4F Fig) or muscle DVB (*lactadherin*, *frem-1*) gene expression (S4G Fig). However, this expression did not form foci or local outgrowths indicative of head formation (S4F Fig). It is possible that anterior PCG expression can be initiated at low levels in the absence of the DVB, and that the DVB acts to promote elevated and sustained anterior PCG expression that culminates in strong foci. Importantly, anterior PCGs were expressed on the uninjured side of both DMSO- and PD-treated animals, where they could form foci, demonstrating that Erk signaling is not generally required for anteriorization during homeostatic turnover (Fig 5B). Together, these results suggest that the DVB is required for robust promotion of anterior patterning and head formation after *β-catenin-1* RNAi.

To further determine whether the DVB is required to promote anterior patterning, we utilized the independent context of head regeneration. Anterior PCGs can become expressed during head regeneration between 24 and 48 hpa [16,44]. Muscle DVB genes became expressed at the edge of head blastemas in a patterned manner by 36 hpa, similar in time to the appearance of anterior PCG foci at the wound (S5A Fig). We utilized inhibition of Bmp signaling to prevent DVB regeneration after transverse amputation. Bmp signaling is required for DV patterning and pathway inhibition can prevent regeneration of the DVB [25–27]. Short-term *bmp4* RNAi tail fragments can express wound-induced genes and rescale positional information despite no DVB regeneration or blastema outgrowth [28]. Interestingly, *bmp4* RNAi has been observed to result in asymmetric poles, displaced laterally [28]. We generated control or *bmp4; smad1* RNAi post-pharyngeal fragments. Control post-pharyngeal fragments expressed anterior PCGs (*sFRP-1*, *ndl-4*) at the midline of the AFW near the regenerated DVB (*laminB*) (Fig 5C). By contrast, *bmp4; smad1* RNAi post-pharyngeal fragments did not regenerate the DVB at their wounds, leaving the fragments with only lateral domains of DVB (pre-existing prior to injury). Anterior PCGs were expressed at the AFW of these *bmp4; smad1* RNAi fragments, but this expression only occurred at the lateral fragment edge near the pre-existing, intact DVB in all cases ($n = 26/26$) (Fig 5C). In some cases, anterior PCGs were expressed on only one side of the fragment at the DVB (left side ($n = 10/26$) or right side ($n = 8/26$)). In other cases, two foci of anterior PCG expression were observed, with one near the left DVB and one near the right DVB ($n = 8/26$).

We next generated *bmp4; smad1* RNAi post-pharyngeal fragments and also removed one side of lateral DVB. Anterior PCGs were expressed at the AFW in most cases ($n = 31/34$; $n = 3/34$, anterior PCGs were not detected). Notably, in these 31 animals, anterior PCG expression was restricted to the lateral edge only, exclusively near the side with remaining DVB ($n = 25/25$, right DVB removal and $n = 6/6$, left DVB removal) (Fig 5C). These results suggest that the DVB is required for sustained anterior PCG expression in regeneration. To assess this possibility, we generated post-pharyngeal *bmp4; smad1* RNAi fragments that also had both sides of the lateral DVB removed. In most cases, *bmp4; smad1* RNAi fragments without any DVB contained few ($n = 7/15$) or no detectable ($n = 6/15$) anterior PCG-expressing cells at 7 days post-amputation (dpa) ($n = 3/15$ had detectable levels of *sFRP-1*, *ndl-4*). These findings support the hypothesis that the DVB is needed for anterior PCG expression in regeneration.

**The DVB has an anterior pole-independent role in the promotion of anterior positional information expression after Wnt inhibition**

The planarian anterior pole coalesces at the DVB [34] and is required to promote sustained anterior PCG expression during head regeneration [31–33]. Perturbations that affect the distribution of DVB cells also alter the location of anterior pole formation [34], suggesting that the DVB plays an active role in its positioning.

In head regeneration, anterior pole formation and anterior PCG expression occur on similar timescales [16,34,44]. However, after *β-catenin-1* RNAi, anterior PCG (*sFRP-1*, *ndl-4*) expression was detected at the DVB prior to anterior pole transcripts being apparent by FISH (Fig 6A). By 7 days of *β-catenin-1* RNAi, 37 ectopic anterior PCG foci were detected across 9 animals. Thirty-three foci across 7 animals and 41 foci across 8 animals were detected at 14 days and 21 days of RNAi, respectively. *FoxD* encodes a Forkhead-family transcription factor that is expressed in a subpopulation of neoblasts (stem cells) during head regeneration and is required for the specification of neoblasts to form anterior pole progenitors [31,32]. Mature anterior pole cells also express *FoxD* [31,32]. Ectopic *FoxD* expression was not reliably detected until day 14 ($n = 10$ foci across 7 animals) to day 21 ($n = 11$ foci across 7 animals) after the initiation of *β-catenin-1* RNAi (Fig 6A). We verified these results using an RNA probe to *notum*, which encodes a secreted Wnt inhibitor expressed in the anterior pole [17,31]. *notum* expression at wounds requires *β-catenin-1* [17]. However, *notum* transcripts could still be detected in the anterior pole after *β-catenin-1* RNAi [19] (S6A Fig). In agreement with the *FoxD* results, ectopic *notum* expression was not detected until day 14 ($n = 1$ focus across 6 animals) to day 21 ($n = 7$ foci across 9 animals) after initiation of *β-catenin-1* RNAi (Fig 6A). Although transcript detection could be limited by FISH sensitivity, this order of events suggests that it is the expression of anterior PCGs at the DVB that is first associated with head formation, and that pole formation might follow and facilitate pattern resolution during head formation.

To test whether the anterior pole is required for anterior PCG expression during homeostatic transformation following *β-catenin-1* RNAi, we inhibited pole formation by *FoxD* RNAi, then fed animals *β-catenin-1* double-stranded RNA (Figs 6B and S6B). *FoxD* inhibition did not dramatically affect anterior PCG (*sFRP-1*, *ndl-4*, *ndl-5*) gradients in control animals (S6C Fig) and *FoxD*; *β-catenin-1* double RNAi animals still induced expression of anterior PCGs at the DVB (Fig 6B), suggesting that the DVB plays an anterior pole-independent role in the promotion of anterior positional information expression after Wnt inhibition.

## Discussion

Positional information along adult planarian body axes is controlled by two largely independent signaling pathways: Wnt for the AP axis and Bmp for the DV axis [10]. This feature is reminiscent of *Drosophila* development, where AP and DV axes are established by largely autonomous, orthogonal patterning systems [62]. Utilization of mechanistically separate molecular pathways to pattern orthogonal tissue axes might represent a common strategy to achieve stable patterns in biology. The planarian AP and DV axes maintain pattern largely independently during homeostatic cell turnover. At the same time, some mechanism of coordination between axes could be deployed in diverse animal contexts to ensure a suitable spatial relationship. Understanding how coordination of tissue axes is achieved is thus fundamental to problems of animal development and regeneration.

We propose that orthogonal coordination of AP and DV patterns in adult planarians occurs primarily during outgrowth formation, such as during head regeneration or homeostatic head formation after Wnt inhibition, as opposed to during pattern maintenance in tissue turnover. Bmp signaling does restrict the posterior extent of Wnt gene expression domains in planarians during tissue turnover [39]. However, we suggest that the primary linkage between patterning of distinct planarian body axis regions during regeneration involves DV-patterning processes that form the DVB at the lateral edge of the DV median plane, or body margin. The DVB then regulates AP-axis growth and patterning, restricting growth and anterior positional information expression to the DVB in a Wnt-low environment (Fig 6C–6E). The linkage of AP-axis patterning

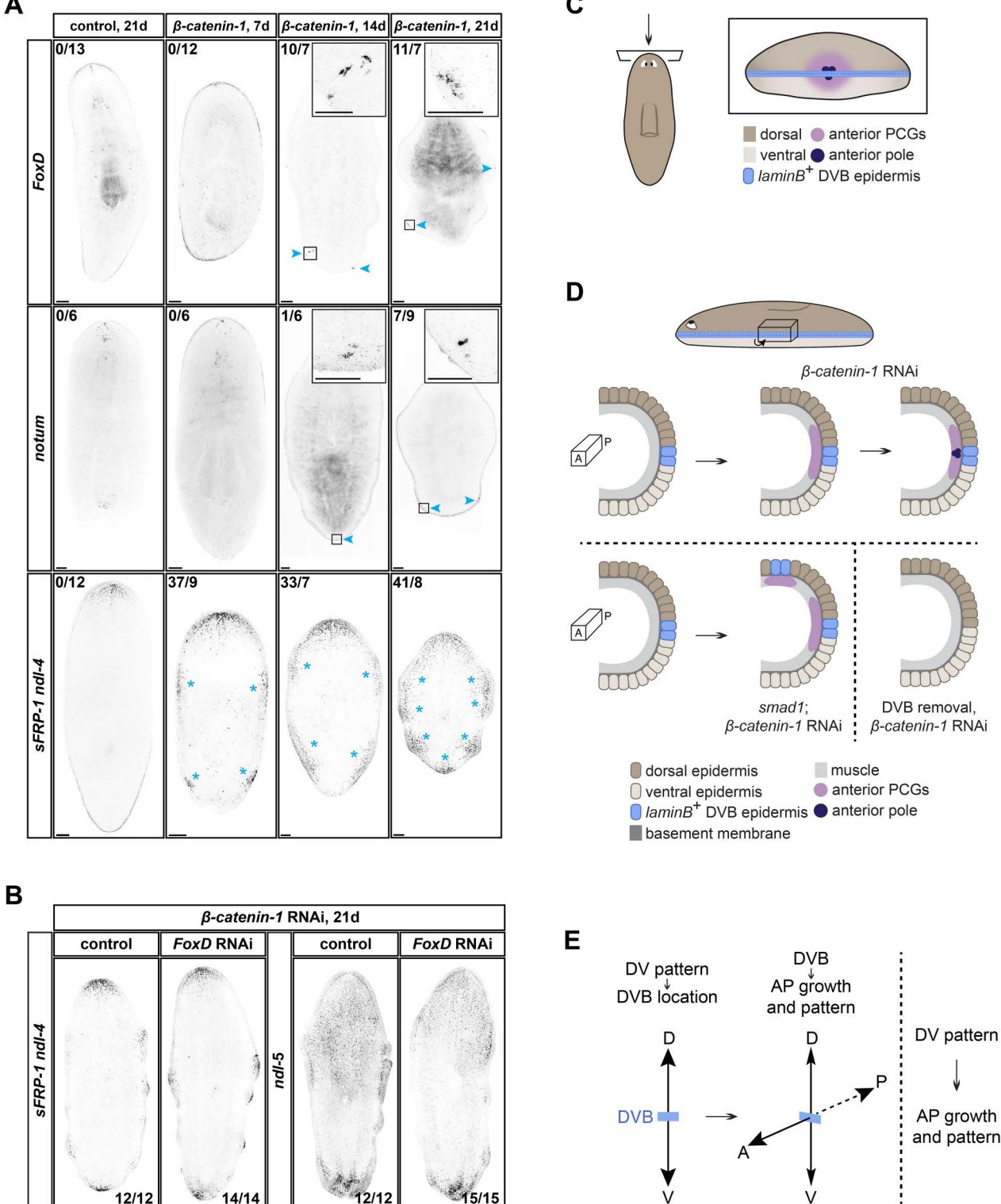

**Fig 6. The DVB has an anterior pole-independent role in the promotion of anterior PCG expression after *β-catenin-1* RNAi. A)** *β-catenin-1* RNAi animals ectopically express anterior PCGs (*sFRP-1*, *ndl-4*) by 7d of RNAi, prior to detection of ectopic anterior pole transcripts (*FoxD*, *notum*) at 14–21d. Blue arrowhead, ectopic anterior pole. Blue asterisk, ectopic anterior PCG focus. **B)** *FoxD; β-catenin-1* RNAi animals express anterior PCGs (*sFRP-1*,

*ndl-4*, *ndl-5*) in ectopic foci at the DVB. **C)** The anterior pole is positioned at the DVB in the head. **D)** Top: Anterior pole formation is preceded by anterior PCG expression at the DVB after *β-catenin-1* RNAi. Bottom left: Ectopic DVB on the dorsal surface of the animal (*smad1* RNAi) leads to the ectopic dorsal expression of anterior PCGs after *β-catenin-1* RNAi. Similar results were obtained after D-V or DVB transplantation (see Fig 3C and 3E). Bottom right: DVB removal (parasagittal cut, PD treatment) blocks lateral anterior PCG expression after *β-catenin-1* RNAi. Similar results were obtained at anterior-facing wounds of *bmp4; smad1* RNAi animals (see Fig 5C). A, anterior. P, posterior. **E)** Model: DV patterning acts to establish the location of the DVB at the midpoint of the DV axis, at the lateral animal margin. The DVB then serves as a landmark to allow growth and patterning of regions of the AP axis, such as during head regeneration or head formation in a low Wnt environment. This allows the DV axis to control orthogonal AP axis region growth and pattern formation. Scale bars, 100 μm (**A**, **B**), 50 μm (**A**, inset).

processes to the position of the DVB is also intimately connected to the ML axis. The anterior pole coalesces specifically at the intersection of three landmarks (the DVB, the midline, and at AFWs) during head regeneration [34]. Transplantation of tissue with reversed DV polarity can lead to the formation of an ectopic head and/or tail in resultant outgrowths [38,63]. Rojo-Laguna (2019) shows that in AP polarized outgrowths the midline marker *slit* becomes patterned along the emergent ML axis, whereas outgrowths that lack AP polarity display aberrant expression of *slit* [63]. It has been proposed that a PAK family kinase coordinates pattern regeneration along the planarian AP and ML axes [64]. After *β-catenin-1* RNAi, ectopic midlines are established in ectopic heads that emerge from the animal lateral edge [19]. Although the role of the DVB in promoting anterior PCG expression after *β-catenin-1* RNAi is independent of the anterior pole, the anterior pole can subsequently coalesce at the DVB, where it likely acts to refine anterior pattern. It is possible that focal anterior PCG expression at the DVB influences the emergence of an anterior pole at the DVB.

Early embryonic patterning can involve influence of one body axis on another. However, border-like signaling centers that are spatially restricted along one axis (such as a planarian DVB-like border) and that influence growth and patterning of another axis are typically not present during initial embryonic axis pattern establishment across animals (e.g., in *Caenorhabditis elegans*, *Drosophila*, zebrafish, *Xenopus*, and mouse). Instead, early pattern-initiating factors (e.g., asymmetric maternal factors, sperm entry location, cortical rotation) lead to patterning processes (such as morphogen gradient formation or organizer formation) that establish axial patterns [65,66]. In *Xenopus* embryos, as in adult planarians, Wnt and Bmp signaling pattern orthogonal AP and DV axes [5–7]. The organizer in *Xenopus* is formed on the prospective dorsal-posterior side of the embryo, following cortical rotation after fertilization, by nuclear β-catenin and vegetal factors (Xnr proteins) [65,67]. There, the organizer controls DV mesoderm patterning and marks the site of gastrulation. This location thereafter expresses Wnt ligands that promote posterior patterning of the AP axis (e.g., AP neuroectoderm patterning) [68]. It has been suggested that integration of Wnt and Bmp signaling can involve Smad1 phosphorylation by GSK3 [69]. Bmp signaling, which regulates DV axis pattern, can also influence convergent extension that promotes lengthening of the AP axis in zebrafish [70].

Patterning centers formed at the midpoint of a tissue axis, however, do have prominent roles in certain later stages of development. For example, the DV boundary in the *Drosophila* wing disc is involved in coordinating growth along the proximodistal axis [71]. In vertebrate limb development, the apical ectodermal ridge resides at the boundary between the dorsal and ventral ectoderm where it promotes proximodistal outgrowth and patterning [72–74]. One possibility is that outgrowths with patterned axes that emerge later in development, such as in appendage formation, or in adulthood, such as during regeneration, are predominant contexts where boundary-like signaling centers have axis-organizing roles. Because these outgrowths need to establish and coordinate axial pattern, they might rely on boundary-like signaling centers with positions determined by embryonic patterning, thereby harnessing existing tissue positional cues to direct new growth. The capacity for whole-body regeneration (ability to form large body axis regions after amputation) is widespread in the Bilateria, including in platyhelminthes, annelids, molluscs, nemertean worms, echinoderms, hemichordates, and urochordates [2]. Whether DVB-like regions exist and promote regeneration in other animals with whole-body regeneration, similar to the case of the planarian DVB, is unknown and will be important to assess. Adult cues, like the DVB, might prove to be fundamental to growth and coordination of axial patterning in regenerative outgrowths broadly in animals.

In addition to DVB-like signaling centers, conflict between dorsal and ventral tissue regions can lead to outgrowth in certain experimental contexts. For instance, DV misalignment, or conflict, generated by transplantation can lead to supernumerary limbs during development [75] and during regeneration [76,77]. Transplantations generating positional conflicts of varied types can lead to regenerative outgrowths in diverse organisms, from crickets [78] to salamanders [79] to planarians [3]. Various models have been proposed to account for these observations, such as those involving intercalation of positional values between juxtaposed positions combined with distalization in the case of certain appendages [80,81] and a boundary model for limbs involving production of distalizing factors at the intersection of AP and DV boundaries [82].

Because of the experimental capacity to induce ectopic DVB and to generate a Wnt-low environment in planarians, unprecedented control over an adult animal body plan can be experimentally achieved. Ectopic heads can be induced to form at desired locations by experimental design. Head shape can also be influenced by the pattern of ectopic DVB, such as being induced to form in goblet-like shapes at DVB rings or along dorsal ridges of DVB. Utilizing a pattern element from one axis to determine the patterning outcome on another axis provides one solution to the substantial challenge in regeneration of replacing tissues with a spatial organization that integrates new tissues with pre-existing patterns in the remaining body.

## Methods

### Animal husbandry

Asexual *Schmidtea mediterranea* strain CIW4 animals were cultured in 1× Montjuic water (1.6 mmol/l NaCl, 1.0 mmol/l CaCl$_2$, 1.0 mmol/l MgSO$_4$, 0.1 mmol/l MgCl$_2$, 0.1 mmol/l KCl, and 1.2 mmol/l NaHCO$_3$ prepared in Milli-Q water) at 20°C in the dark. Animals were fed homogenized beef liver and starved at least 7 days prior to experiments.

### 10× single-cell mRNA sequencing and analysis

Tissue fragments from the lateral edges of ~20 large animals were amputated using a scalpel and collected in 1× Montjuic water. Montjuic water was replaced with 0.25% Trypsin-EDTA (1×) and tissue fragments were incubated for 6 min with vigorous pipetting to allow for cell dissociation. The sample was then centrifuged at 500$g$ for 5 min, resuspended in ice-cold CMFB [calcium–magnesium free solution with 0.1% bovine serum albumin (BSA) (400 mg/L NaH$_2$PO$_4$, 800 mg/L NaCl, 1,200 mg/L KCl, 800 mg/L NaHCO$_3$, 240 mg/L glucose, 0.1% BSA, 15 mM HEPES, pH7.3)], and passed through a 40 µm filter. Debris was removed from the dissociated cells with a spiral microfluidic device. The device was a gift from Kyungyong Choi and Jongyoon Han at MIT. It was used as described in [83]. Briefly, the cells were suspended in 20 mL of CMFB and circulated through the device at 16 mL/min in closed-loop separation mode, returning the output of the inner wall outlet to the sample tube, and discarding the output from the outer wall outlet. Once the volume in the sample tube was reduced to approximately 5 mL, the sample was washed at the same pump rate with an additional 40 mL of CMFB, which was added 1 mL at a time to the sample tube so as to keep the sample volume at approximately 5 mL. After washing, the sample was concentrated by further pumping until the input tube was nearly empty. The sample input tube was then disconnected and the sample (approximately 3 mL) was collected by directing both the outer and inner outlet tubes to a sample collection tube and pumping at 1 mL/min until the device was empty. After sorting, the sample was stained with trypan blue to count viable cells for the 10× single-cell mRNA sequencing procedure. Cells were processed by the Whitehead Institute Genome Technology Core (WIGTC) using 10× Genomics Chromium Controller and Chromium Single Cell 3′ Library & Gel Bead Kit following standard manufacturer's protocol. The sample was sequenced on a NovaSeqSP (100×100 paired-end reads). Sequencing reads were mapped using a GTF file of Smed_v6 (https://planmine.mpinat.mpg.de/planmine/model/bulkdata/dd_Smed_v6.pcf.contigs.fasta.zip) genes in the context of the Smes_g4 (https://planmine.mpinat.mpg.de/planmine/model/bulkdata/dd_Smes_g4.fasta.zip) genome. This GTF file was generated by using BLAT to map all Smed_v6 transcripts to the Smes_g4 genome and each transcript was assigned to a single genome

location based on the best alignment score. Transcripts were then collapsed using genome location prior to mapping using the Cell Ranger 7.2.0 pipeline. Cells were assessed for nUMI, nGene, and percent mitochondrial transcript content, which were represented in violin plots. Percent mitochondrial content was based on mitochondrial genes reported in [28] which are represented in v_6 of the Dresden transcriptome (dd_Smed_v6_258_0_1, dd_Smed_v6_289_0_1, dd_Smed_v6_292_0_1, dd_Smed_v6_297_0_1, dd_Smed_v6_344_0_1, dd_Smed_v6_505_0_1, dd_Smed_v6_753_0_1, dd_Smed_v6_957_0_1) and on the highly abundant mitochondrial transcripts (mtRNA_1, mtRNA_2) from [84]. Any cells with nFeature_RNA < 500 or nFeature_RNA > 4,000, or nCount_RNA < 1,000 or nCount_RNA > 10,000, were removed from the dataset prior to analysis. Doublets were identified using scDblFinder (https://bioconductor.org/packages/release/bioc/html/scDblFinder.html) and removed after basic QC filtering. 10× analysis was performed using Seurat 5.1.0 where cells were visualized using the uniform manifold approximation and projection (UMAP) algorithm. The number of dimensions used with RunPCA, RunUMAP, and FindNeighbors was determined using JackStraw with a $p$-value cutoff of 0.05. Clusters were determined via FindClusters using the Leiden algorithm. UMAP plots of identities were created using Seurat's DimPlot function. UMAP plots of gene expression were created using Seurat's FeaturePlot function. Positively enriched genes were identified using Seurat's FindMarkers function. Human best BLASTx hits were identified using the Human GENCODE Gene Set v39.

## RNAi

For RNAi experiments, double-stranded RNA (dsRNA) was synthesized by in vitro transcription reactions (Promega) using PCR-generated templates with flanking T7 promoters, followed by ethanol precipitation, and annealed after resuspension in water. The concentration of dsRNA varied in each prep between 4 and 7 μg/ml. dsRNA was mixed in a 1:2 ratio with liver and 1−2 μl of this mixture (liver containing dsRNA) per animal was used for feedings. *C. elegans unc-22* was used as the control condition. Homeostatic *β-catenin-1* RNAi animals were fed three times in 2 weeks and fixed at 21 days (Figs 1A, S1A, and S1C). For S1B Fig, *β-catenin-1* RNAi animals were fed twice and fixed after 7 days. Homeostatic *bmp4* RNAi animals were fed six times in 3 weeks, then once per week until fixation at 45 days (Figs 1B and S1D–S1G). For epidermal DVB FISH, muscle fiber type FISH, and muscle DVB FISH, *β-catenin-1* RNAi animals were fed three times and fixed at 14−21 days (Figs 1E—1G and S1L–S1N). Transplant animals were fed *β-catenin-1* dsRNA once 10−14 days after surgery and fixed 10−14 days later (Figs 3A—3E and S3D). *smad1* RNAi animals were fed 1-part *smad1* dsRNA mixed with 1-part control dsRNA. After 2−3 weeks, when animals had developed dorsal DVB ridges, animals were fed *β-catenin-1* dsRNA once and fixed 10 days later (Figs 4A and S3E). For panel 4B, RNAi was administered as described in [85] and animals were fixed after 77d of *smad1* RNAi and 21d of *β-catenin-1* RNAi. For *smad1* RNAi dorsal injury experiments, animals were fed 1-part *smad1* dsRNA mixed with 1-part control dsRNA and injured 3 days later. Animals were fed *β-catenin-1* dsRNA 10 days following injury and fixed 12 days after that (Fig 4C and 4D). For lateral versus internal injury experiments, *β-catenin-1* RNAi animals were fed once, injured 3 days later, and fixed 3 days after injury (Figs 5A, S4A, and S4B). For most Erk inhibition experiments, *β-catenin-1* RNAi animals were fed once, injured 3 days later, and fixed 7 days after injury (Figs 5B, S4C, S4D, and S4F). For the muscle DVB FISH of *β-catenin-1* RNAi, PD-treated animals, animals were fed with *β-catenin-1* dsRNA twice, injured 5 days after the initiation of RNAi, and fixed 5 days after injury (S4G Fig). For the homeostatic Erk inhibition experiment, animals were fed *β-catenin-1* dsRNA twice, incubated in PD immediately following the first feeding, and fixed 10 days after the initiation of RNAi (S4E Fig). *bmp4; smad1* RNAi animals were fed four times in 2 weeks. Animals were amputated 7 days after the last feeding and fragments were fixed 7 days after amputation (Fig 5C). For timecourse experiments, homeostatic *β-catenin-1* animals were fed two (7 days) to three (14 days, 21 days) times and fixed 5−14 days after the last feeding (Figs 6A and S6A). *FoxD* RNAi animals were fed six times in 3 weeks. *β-catenin-1* dsRNA was then fed two times in 1 week and animals were fixed after 21 days (Figs 6B, S6B, and S6C).

## Transplantation procedures

Animals were soaked in 0.2% chloretone for 2 min and placed on moist WhatmanTM filter paper (GE Healthcare, Life Sciences) on a cold block to limit movement. A ~250 µm piece of tissue at the midline posterior to the pharynx was excised from a recipient animal, and the animal was allowed to recover in Holtfreter's solution for 2 min. For D-V transplants, after a second incubation in chloretone, donor tissue (from a matched AP and ML location) was transplanted into the recipient with reversed DV polarity. For DVB transplants, a ~250 µm piece of tissue from the donor animal lateral edge (at a matched AP location) was transplanted into the midline posterior to the pharynx of a recipient. The recipient animal was then covered with cigarette rolling paper soaked in Holtfreter's solution and placed at 10°C in the dark. After 24 hours, animals were placed in planarian water and returned to 20°C.

## Erk signaling inhibition

PD0325901 (in short, PD) was dissolved in DMSO, used at 10 µM, and replaced daily. Animals were fed *β-catenin-1* dsRNA, incubated in PD immediately following feeding, and amputated 3–5 days later. Animals were fixed for further analysis 5–7 days after parasagittal amputation.

## Fluorescence in situ hybridizations

RNA probes were synthesized in vitro and whole-mount FISH was performed. Briefly, animals were killed in 5% NAC and treated with proteinase K (2 µg/ml). Following overnight hybridizations, samples were washed twice in pre-hybridization buffer, 1:1 pre-hybridization-2 × SSC, 2 × SSC, 0.2 × SSC, PBS with Triton-X (PBST). Subsequently, blocking was performed in a 10% Western Blocking Reagent (Roche, 11921673001) PBST solution for DIG and DNP probes, or in a 5% Western Blocking Reagent and 5% Horse serum PBST solution for FITC probes. Samples were incubated overnight at 4°C in antibody. Antibody washes were then performed for one hour, followed by tyramide development. Peroxidase inactivation with 1% sodium azide was done for 90 min at room temperature.

## Microscopy and image analysis

Brightfield images were taken with a Zeiss Discovery Microscope. Fluorescent images were taken with a Leica SP8 or Leica Stellaris confocal microscope. Fiji/ImageJ and Adobe Photoshop were used to perform brightness and contrast adjustments. All FISH images shown are representative of all images taken in each condition and are maximum intensity projections. PCG domain lengths were quantified in a condition-blind manner using FIJI/ImageJ. A line was drawn at the posterior or anterior boundary of prominent PCG signal and the distance from this line to the head or tail tip was measured and normalized to animal length. To generate line plots of signal intensity, fluorescence intensity was measured along a line of width 100 pixels drawn through the midline from the dorsal to ventral side of the animal using FIJI/ImageJ. Raw data was smoothed using a Gaussian filter.

## Supporting information

**S1 Fig. Validation of *β-catenin-1* and *bmp4* RNAi phenotypes. A)** Posterior-lateral expression of anterior PCGs (*sFRP-1*, *ndl-4*) following *β-catenin-1* RNAi (18–21d). Foci containing multiple anterior PCG+ cells were apparent at the DVB (foci typically had >10 cells). Scattered anterior PCG+ cells were also present in the animal. **B)** Ventrally biased (*admp*, *nlg-7*) PCG expression at the DVB after *β-catenin-1* RNAi (7d). C) Dorsally biased (*bmp4*, *nlg-8*) and ventrally biased (*admp*, *nlg-7*) PCG expression in tails after *β-catenin-1* RNAi (21d). **D)** Dorsal expression of the ventral neural marker *eye53-1* following *bmp4* RNAi (45d). **E, F)** PCG expression domains in *bmp4* RNAi animals (45d). Blue arrow denotes anterior (posterior PCG) or posterior (anterior PCG) boundary of PCG expression domain. **G)** The posterior expansion of *notum* signal on the dorsal surface after *bmp4* RNAi is because of expansion of the anterior pole

(*notum*+*ChAT*−). **H)** Fluorescence intensity of the PCGs *bmp4* and *admp* and the epidermal DVB marker *laminB* along a line is plotted. Smoothed data is represented with a thick stroke. Individual data points are listed in S1 Data. **I)** Anterior and posterior PCG expression is concentrated near the DVB during regeneration (96 hpa). Single-channel images of the merged images in Fig 1C are presented. **J, K)** Fluorescence intensity of anterior and posterior PCGs and the epidermal DVB marker *laminB* along a line is plotted. Smoothed data is represented with a thick stroke. Individual data points are listed in S1 Data. **L)** Ectopic anterior PCG (*ndl-4*) expression near the DVB after *β-catenin-1* RNAi (14d). D, dorsal; V, ventral. Fluorescence intensity along a line is plotted and individual data points are listed in S1 Data. **M)** Ectopic anterior PCG (*sFRP-1*, *ndl-4*) expression after *β-catenin-1* RNAi (18d) is detected in longitudinal (*myoD*+), circular (*nkx1-1*+), and lateral DV (*nk4*+) muscle at the lateral animal edge. Single-channel images of the merged images in Fig 1F are presented. **N)** Co-expression of anterior PCGs (*sFRP-1*, *ndl-4*) and the secreted molecules *lactadherin* and *netrin-1* in muscle at the DVB after *β-catenin-1* RNAi (18d). Colored boxes, area depicted in photos. Scale bars, 100 μm (A, C–G), 50 μm (B, I, L–N), 25 μm (G, inset).
(PDF)

**S2 Fig. Identification of novel DVB marker genes. A)** Cartoon shows region of the animal isolated for scRNA-seq. Violin plots show the number of UMIs (left) and genes (center) per cell. Individual data points are listed in S1 Data. Table (right) shows the total number of cells after quality control (QC), mean reads, and mean genes detected per cell. **B)** Violin plot shows the percentage of reads mapped to mitochondrial genes. Individual data points are listed in S1 Data. **C)** Sub-clustering of muscle cells, labeled by *colF-2*, reveals the muscle subtype composition at the DVB. UMAP plots show the overlapping expression of DVB marker genes in longitudinal muscle cells, labeled by *myoD*. Novel markers of the DVB are labeled in black. Genes with DVB expression verified by FISH are separated by dotted lines. **D)** UMAP plots show the expression of DVB marker genes in circular muscle cells. **E)** Heatmap shows the expression of FISH-screened subcluster 1-enriched genes across *myoD*+ longitudinal muscle subclusters. **F)** UMAP plots show DVB-enriched gene expression. **G)** Table of 32 longitudinal muscle subcluster 1-enriched genes categorized by the type of protein each gene encodes. Genes labeled in bold font have DVB expression patterns verified by FISH. **H)** Table of 11 unnamed DVB-enriched genes listed by dd_ID. The PFAM domains comprising proteins encoded by these genes are listed. **I)** Protein domain structure of novel DVB markers. **J)** Whole-body expression of novel DVB marker genes by FISH. **K)** Plot shows the Pearson correlation coefficients for pairwise comparisons between DVB-enriched genes in longitudinal muscle subcluster 1. Scale bars, 100 μm (J).
(PDF)

**S3 Fig. Controls for ectopic DVB experiments. A)** D-V transplant animals one day post-transplant (dpt). **B)** D-V transplant outgrowths show muscle DVB gene expression (*lactadherin*, *frem-1*). **C)** Post-pharyngeal D-V transplant outgrowths express *wntP-2*, indicative of posterior identity. **D)** Ventral outgrowths that result from D-V transplant are anteriorized (*sFRP-1 ndl-4*+) after *β-catenin-1* RNAi (10–14d). **E)** *smad1* RNAi animals retain their original DVB (*laminB*), which is anteriorized (*sFRP-1 ndl-4*+) after *β-catenin-1* RNAi (10d). Yellow arrowhead, anterior PCG focus. Scale bars, 250 μm (A), 100 μm (B–E).
(PDF)

**S4 Fig. Controls for DVB injury and removal experiments. A)** Lateral wedge and internal hole punch animals one day post-injury (dpi). B) *β-catenin-1* RNAi animals after internal hole punch. In rare cases (*n* = 7/59), a focus of anterior PCG (*sFRP-1*, *ndl-4*) expression develops at the wound. *β-catenin-1* RNAi, hole punch animals that do not ectopically express anterior PCGs at the wound (*n* = 51/59) are shown for comparison and the image from Fig 5A is presented. **C)** *β-catenin-1* RNAi animals incubated in PD one dpa. **D)** PD treatment blocks regeneration of the DVB (*laminB*) following parasagittal amputation (*n* = 18/21). In 3/21 animals, a patch of *laminB*+ DVB was present after parasagittal amputation and PD treatment, indicating incomplete DVB removal or partial DVB regeneration. **E)** PD treatment does not prevent homeostatic

outgrowth associated with ectopic head formation after *β-catenin-1* RNAi (10d). **F)** *β-catenin-1* RNAi animals incubated in PD after parasagittal amputation. In some animals (*n* = 9/22), anterior PCG (*sFRP-1*, *ndl-4*) expression is detected at the wound, though these animals lack PCG foci formation and apparent homeostatic outgrowth. *β-catenin-1* RNAi, PD-treated animals that lack anterior PCG expression at the wound (*n* = 12/22) are shown for comparison and the image from Fig 5B is presented. One animal had a patch of *laminB*+ DVB present at the wound (*n* = 1/22), indicating incomplete DVB removal or partial DVB regeneration. **G)** PD treatment after DVB removal blocks regeneration of muscle DVB (*lactadherin*, *frem-1*) cells. Despite no DVB regeneration, low levels of anterior PCG (*sFRP-1*, *ndl-4*) expression can be detected at the wound (*n* = 9/10). Colored boxes, area depicted in photos. Scale bars, 250 μm (A, C, E), 100 μm (B, D, F, G), 50 μm (F, inset), 25 μm (G, inset).
(PDF)

**S5 Fig.** Muscle DVB gene expression regeneration. **A)** Regeneration of anterior PCG (*sFRP-1*, *ndl-4*) and muscle DVB (*lactadherin*, *frem-1*, dd_30210, *egf-6*) gene expression domains 0–72 hpa. Patterned muscle DVB gene expression is detected in the blastema by 36 hpa, along with a focus of *sFRP-1 ndl-4* expression. Colored box, area depicted in photos. Scale bars, 50 μm (A).
(PDF)

**S6 Fig.** *FoxD* **RNAi blocks anterior pole formation. A)** Detection of anterior pole transcripts (*FoxD*, *notum*) after *β-catenin-1* RNAi (7–21d). **B)** *FoxD* expression in the anterior pole is significantly reduced (*n* = 6/12) or absent (*n* = 6/12) after *FoxD; control* RNAi and absent (*n* = 12/12) after *FoxD; β-catenin-1* RNAi. **C)** Anterior PCG (*sFRP-1*, *ndl-4*, *ndl-5*) gradients are not grossly affected by *FoxD* RNAi. Colored boxes, area depicted in photos. Scale bars, 50 μm (A, B), 100 μm (C).
(PDF)

**S1 Table.** **Subcluster-enriched genes identified by DVB scRNA-seq data.**
(XLSX)

**S2 Table.** **Contig annotation of genes used in figures.**
(XLSX)

**S1 Data.** **Data used to generate plots.**
(XLSX)

## Acknowledgments

The authors thank members of the Reddien lab for helpful comments and discussion. We thank Conor McMann for measuring PCG gradient lengths in *bmp4* RNAi animals and Patrick Aoude for 10× mapping and computational support.

## Author contributions

**Conceptualization:** Chloe L. Maybrun, Isaac M. Oderberg, Michael A. Gaviño, Peter W. Reddien.

**Data curation:** Chloe L. Maybrun, Peter W. Reddien.

**Formal analysis:** Chloe L. Maybrun, Peter W. Reddien.

**Funding acquisition:** Peter W. Reddien.

**Investigation:** Chloe L. Maybrun, Isaac M. Oderberg, Michael A. Gaviño, Peter W. Reddien.

**Methodology:** Chloe L. Maybrun, Thomas F. Cooke, Kyungyong Choi, Jongyoon Han, Peter W. Reddien.

**Project administration:** Peter W. Reddien.

**Resources:** Peter W. Reddien.

**Supervision:** Peter W. Reddien.

**Validation:** Chloe L. Maybrun, Peter W. Reddien.

**Visualization:** Chloe L. Maybrun, Peter W. Reddien.

**Writing – original draft:** Chloe L. Maybrun, Peter W. Reddien.

**Writing – review & editing:** Chloe L. Maybrun, Isaac M. Oderberg, Peter W. Reddien.

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
