## [Editor Report · Decision Letter 0]

11 Feb 2025

Dear Dr Reddien,

Thank you for submitting your manuscript entitled "The dorsal-ventral boundary influences anterior-posterior axis formation in planarians" for consideration as a Short Report by PLOS Biology.

Your manuscript has now been evaluated by the PLOS Biology editorial staff, as well as by an academic editor with relevant expertise, and I am writing to let you know that we would like to send your submission out for external peer review.

Once your full submission is complete, your paper will undergo a series of checks in preparation for peer review. After your manuscript has passed the checks it will be sent out for review. To provide the metadata for your submission, please Login to Editorial Manager (https://www.editorialmanager.com/pbiology) within two working days, i.e. by Feb 13 2025 11:59PM.

Kind regards,

Richard

Richard Hodge, PhD

rhodge@plos.org

PLOS

---

## [Decision Letter · Decision Letter 1]

8 Apr 2025

Dear Dr Reddien,

Thank you for your patience while your manuscript "The dorsal-ventral boundary influences anterior-posterior axis formation in planarians" was peer-reviewed at PLOS Biology as a Short Report. Please accept my sincere apologies for the delays that you have experienced during the peer review process. Your manuscript has now been evaluated by the PLOS Biology editors, an Academic Editor with relevant expertise, and by three independent reviewers.

In light of the reviews, which you will find at the end of this email, we would like to invite you to revise the work to thoroughly address the reviewers' reports.

As you will see, the reviewers are generally positive about your study and think it is well done and presented. Reviewer #1 raises concerns with the conceptual framing of the study and notes that the manuscript should be refocused on the role of the DV border as a signaling center. The reviewer also notes several cases where the conclusions are not adequately supported by the data. In addition, Reviewer’s #1 and #2 note that the DVB markers identified in the scRNA-seq dataset were not used to provide a more in depth analysis of their signaling roles at the DV border. After discussions with the Academic Editor, we encourage you to strengthen the study along these lines and use the gene expression dataset to provide some preliminary insights into their roles in DVB formation or maintenance.

Given the extent of revision needed, we cannot make a decision about publication until we have seen the revised manuscript and your response to the reviewers' comments. Your revised manuscript is likely to be sent for further evaluation by all or a subset of the reviewers.

**IMPORTANT - SUBMITTING YOUR REVISION**

*Re-submission Checklist*

*Published Peer Review*

*PLOS Data Policy*

*Blot and Gel Data Policy*

Sincerely,

Richard

Richard Hodge, PhD

rhodge@plos.org

REVIEWS:

Reviewer #1: In the manuscript entitled 'The dorsal-ventral boundary influences anterior-posterior axis formation in planarians' Maybrun et al. study the role that the DV border could have in allowing the emergence of the AP axis. For doing that they promote the induction of ectopic DV borders, or supress its formation, and they observe a correlation between the existence of a DV border and the expression of anterior markers (anterior PCGs). The authors also analyse at single cell level the DV border and identify different cell clusters. The authors conclude that there is a temporal and hierarchical relationship between the appearance of the DV border and the formation of the AP axis.

The study demonstrates the importance of the DV border in promoting the growth of a new axis. The essential role of the DV border as a region with the competence to sustain the emergence of a new axis is not new. However, its demonstration in the specific context of planarians and whole-body regeneration—through such an original approach (inducing ectopic DV borders in combination with bcatenin1 inhibition)—is elegant, novel, and highly informative. Additionally, the SC-seq analysis of this specific region provides valuable data for the field.

However, there are major issues regarding some sections of the manuscript and the conclusions that need to be addressed.

Major concerns

1) The more important and general concern is that the requirement of a DV confrontation to promote the growth of a new axis is a well-established mechanism in the field of developmental biology. For instances, the experiments in cricket or axolotl limbs in which grafting a rotated limb promotes the growth of ectopic limbs are well known. The fact that to trigger the regenerative response in planarians it is required the close of the wound and thus the contact between the D and the V domains is also well established. For this reason, 1) this background should be included in the introduction and also in the discussion, and 2) all the data presented which refers to the DV axis is in fact related to the DV border (generated by the confrontation of the D and V domain). So, it does not refer to an axis but to a specific domain that acts as a signalling center (in all metazoans).

Focusing in planarians, after an amputation, the D and V domains get in touch in the wounded area, and this confrontation of signals triggers the regenerative response, which includes the expression of wound response genes, and PCGs (AP, DV and ML). It is true that the first event is the DV confrontation and thus the emergence of the DV border, but the DV axis was already present in this first stage, when new tissue is still not present. Even more, in the current study the process of regeneration in a wt is not studied, but only the expression of A PCGs in ectopic DB borders (which already have an established DV axis). For this reason, the conclusion illustrated in figure 5E is confusing. In this study it is not demonstrated that there is a hierarchical relationship between both axes, but that A identity (not an AP axis, because to define an axis 2 poles are required) can only be generated in a DV border region. Furthermore, that 'this mechanism enables coordination of orthogonal positional information in the context of regeneration' cannot be concluded in this study, since the orthogonal relationship between bot axis is not studied at all.

In summary, the conceptual framework of the study is not the establishment and coordination of the DV and AP axes but rather the role of the DV border as a signaling center or a permissive context in which a second axis can grow. This requires a refocusing of all sections in the manuscript.

2) The results and conclusion of the first section are not supported and furthermore they are not relevant for the rest of the findings of the manuscript. Even the meaning of the title is confusing ('Positional information along the AP and DV axes is largely independently maintained'). If the message of this section is that during homeostasis the PCGs are stably maintained, in particular the DV genes in bcatenin1 RNAi animals, this is not observed in Figure 1A. In this figure, bmp4 and ngl-7 expression in the DV border of bcatenin1 RNAi animals is different than in controls. In bcatenin1 RNAi animals bmp4 it's seen in the V view and in controls in the D view. ngl-7 expression in the DV border is not seen in bcatenin1 RNAi animals, but it is in controls.

3) While the experiments in which an ectopic DV border is induced do show that A PCGs only appear in these regions, the conclusions after ERK and DV border suppression are not so informative. It is reported in several studies that ERK inhibition blocks the wound response and subsequently any regenerative response. Thus, in the context of ERK inhibition no DV border is generated, because as explained few lines above, the appearance of the DV border signalling center is linked to the regenerative response in general (wound response genes, PCGs of all axis).

In any case, it is very surprising that even when ERK is not signalling, 9/22 animals show A PCGs expression (see a further comment below). And then in line 239, it is indicated that those regions positive for A PCG do not form overgrowths. This is not indicating that the DV border is required for the growth of an axis, but that ERK blocks any regenerative response, so an overgrowth cannot occur in these conditions.

The conclusion that 'Removal of the DVB region blocks anteriorization caused by Wnt inhibition or anterior positional information activation in the Wnt-low environment of head regeneration', although it is true, it is misleading, because removal of ERK and the DV border blocks any regenerative signal.

4) The relevance of the information obtained after the SC-seq analysis of the DV border is not clear. The different cell populations and the markers identified could be used to analyse more in deep the signalling properties of the DV border. For instances, even when ERK is not signalling, 9/22 animals show A PCGs expression. This result contradicts authors hypothesis, and deserved more attention. Here it is an opportunity to check if in these animals, despite not having laminb, they show expression of other DV markers that could be indicating the presence of a functional DV border.

5) Last section 'The DVB has an anterior pole-independent role in the induction of anterior positional information expression after Wnt inhibition' is very confusing. It is published by the same authors of this manuscript that notum expression is downstream of bcatenin1. So, bcatenin1 RNAi animals do not show expression of notum although they regenerate heads. Some days after the RNAi inhibition of bcatenin1 expression of notum can appear, since the dsRNA is not present anymore. This is what could be happening in the animals in Figure 5A.

So, the conclusion that the 'DV border plays an anterior pole-independent role in the induction of anterior positional information expression' could be replaced by 'bcatenin1 plays an anterior pole-independent role in the induction of anterior positional information expression', which is another way of saying what was already published by the same authors, that a head can be formed in absence of notum (anterior pole) if bcatenin1 is silenced.

It could be that notum is required to inhibit bcatenin1 when an A pole must be specified, but in a very low Wnt environment this is not necessary anymore, and a head is formed, in the absence of the anteror-pole determinant notum. The role of the DV border in this process is not known. May be silencing of specific markers of the DV border (from the SC-seq data for instances) could be useful to identify specific factors of the DV border which confer properties as a competent region to promote A PCGs expression.

Minor concerns

1) In the SC-seq analysis, laminB was used to assign a D or V position. Why laminB is considered the reference of the DV border? Could it be that laminB is in fact in the D or V domain and other genes of the atlas are located in the physical position of the cells in the DV border?

2)The existence of 2 axis is linked to the appearance of the midline. If the study aims to analyse the relationship between the DV and the AP axis (which is not the current form), considering the midline is essential. Not only during regeneration of a wt animal (Oderberg et al. 2017), also in bcatenin1 RNAi animals each ectopic head always show expression of slit in the midline (Stückemann et al. 2017). Rojo-Laguna et al. 2019 (a missing reference) also show that all overgrowths have DV border (if-b) but they only have AP identity when they recover the midline (slit).

Another context that demonstrates the requirement of the midline in addition to the DV border is that when a wound is generated near the head of a wt animal, which has already a low Wnt environment, it does not promote the expression of A PCGs and the formation of an ectopic head. In wt animals it only occurs in the DV border that crosses the midline region.

The analysis or at least a in deep discussion on the role of the midline is required.

3) When performing grafts that induce new DV confrontations there appear overgrowths, which can have different shapes and identities. For instances, Rojo-Laguna et al. 2019 show that the inclusion/exclusion of the VNCs gives rise to different overgrowths shape and identities. It is not clear in the present manuscript how the grafts were performed (a more detailed procedure is required) neither the identity of the outgrowths in the different contexts.

In lines 184-185- the authors describe that the outgrowths in Figure 3 have the AP character typical of the location of the transplant. Does it mean that if they are posterior to the pharynx, they are P, and if they are in the prepharyngeal region they are A? Then, the overgrowth in Figure 3C has P identity? This should be demonstrated with the corresponding markers if it is stated in the text. In addition, the outgrowths produced after the DV confrontation in control and bcatenin1 RNAi animals appear in D and in V regions. Only the D ones have A identity in bcatenin1 RNAi animals? What about the identity of the V outgrowths? The A and P identities of the outgrowths should be clarified with markers when required as well as a better explanation of the procedures and results.

4) The analysis of the expression of DV and AP (and ML) PCGs in controls compared to different RNAi conditions would be very informative in regenerating animals if the purpose is to understand the hierarchy of the body axis during regeneration. For instances, do DV or DV border genes appear earlier than AP genes? How does the RNAi of DV genes affects the expression of AP genes? And the other way around?

Reviewer #2: The manuscript provides novel evidence that the dorsoventral boundary (DVB) in planarians is necessary and sufficient for the expression of anterior genes and subsequent head formation, especially under conditions that elicit the formation of ectopic heads via reduced canonical Wnt signaling. The authors use both surgical and molecular manipulations to create or eliminate DVBs and show that these can act as foci for anterior gene expression. They further provide evidence that anterior gene expression does not rely upon the formation of anterior pole cells. While the study does not investigate the molecular mechanisms linking the DVB to anterior gene expression, the data is largely sound and well-presented.

The authors also uncover new genes expressed along the DVB which will be valuable for future studies. But the new DVB genes were not utilized as markers in this study, and their roles in DVB formation or maintenance were not tested.

I do have a few concerns on specific results, or on the presentation of results, that I would like to see addressed.

1) Anterior markers after beta-catenin-RNAi. The authors suggest throughout the manuscript that there is a tight correlation between the DVB and the ectopic expression of anterior makers after knockdown of canonical Wnt signaling by beta-catenin-RNAi. However, in Figure S1A the ectopic sFRP-1 induced by the RNAi was not limited to the DVB. Although it was stronger in foci near the DVB, there was also scattered ectopic expression throughout the animal that appears every bit as strong as that near the head. I think this needs to be briefly discussed, and the authors need to be clear on what level and concentration of expression they are referring to elsewhere in the manuscript.

2) "111 In agreement with findings in Clark, 2023 [26], we found that the domain of

112 wnt1+ cells (the posterior pole) was expanded after 45 days of bmp4 RNAi (Figure 1B,

113 Figure S1D)."

To my eye the expression looks ectopic rather than expanded, as it is largely limited to a mid-posterior region that does not extend to the domain of normal expression at the posterior pole. However, this statement, and the bars in Fig. 1B, imply continuous expression from the posterior to mid-posterior.

3) "236 In some cases (n=9/22), reduced anterior PCG

237 (sFRP-1 and ndl-4) expression was detected on the injured side of PD-treated β-

238 catenin-1 RNAi animals, despite no regeneration of laminB cells at the DVB."

The statement is confusing: "reduced" compared with what? I think the authors mean "low-level expression was detected".

4) Fluorescence images and PD treatment- The authors show multicolor fluorescence images that are, on my screen, rather dark, and the overlapping domains of expression are never presented as single-channel images. This makes it very difficult to judge the purple channel anterior marker data, either when it is low or when it overlaps strong green staining. I think it is important to show the single channel data on its own.

The issue is especially critical for the PD treatment data of Figures 4B and S5D-E. First, when viewed without the green channel Fig S5D shows that MEK inhibition via PD treatment alone greatly increases anterior gene expression in all the remaining DVB cells. If this is not a mistake with the authors' image processing this is an important (and confounding) result that the authors need to address. The authors are using DB treatment as way of blocking regeneration of the DVB after surgical removal, and in 4B and S5E say this largely blocks expression of anterior genes after beta-catenin-RNAi in the region where the DVB was removed. However, if DB treatment on its own causes anterior-gene expression along the DVB, then the RNAi experiment seems superfluous.

Additionally, the authors claim to see two different classes of results in S5E, but the differences in intensity of the anterior genes on the left DVB of each figure are not that obvious when viewed without the green channel.

Minor note: placing the 12/22 over the yellow detail box in 4D, and then having the magnified version of the box say 22/22, is confusing.

5) The section about the bmp4;smad-1 RNAi experiments could be clarified. The 3/9 vs 6/9 of Fig. 4C is never referred to in the text or legend. The "By contrast" sentence about the lack of regeneration of the DVB from the RNAi-treated fragments seems out of place, as the next two sentences refer to fragments that had their DVBs intact. And for:

"In bmp4;smad-1 RNAi fragments with

262 one DVB side removed, anterior PCGs were expressed at the AFW in most cases

263 (n=9/10). However, anterior PCG expression was restricted to the lateral edge only near

264 the remaining DVB in all cases (n=9/9) (Figure 4C). "

the authors do not say what happens in the 1/10 fragment; no PCG expression? And is the last sentence referring to the 9 of that 10 that had the DVB removed, or to the 9 with the DVB intact?

Reviewer #3: This is a very nice report that advances our understanding of the relationship between D/V patterning and A/P specification in Planaria. It builds on previously published work from the Agata lab from a quarter of a century ago, as well as more recent work indicating a central role for BMP/SMAD signaling in controlling the D/V axis. While it was known previously that ectopic head formation caused by RNAi of beta-catenin occurs at the lateral periphery of these Planaria, the authors use scRNAseq to define these cells at the D/V boundary (DVB). They go on to show through a combination of grafting (ala Agata and colleagues) and compound RNAi experiments (making very nice use of the temporal control of administering the dsRNA - a feature that might be unique to Planaria) that the DVB cells are sufficient and necessary for ectopic head formation. They also do a careful temporal analysis to support a model where DVB expression at the site of ectopic heads precedes that of the anterior pole.

I can't identify any significant weakness in this story. Also, I believe this is a significant enough advance in our understanding of how two axes coordinate during regeneration to merit publication in PLOS Biology.

---

## [Decision Letter · Decision Letter 2]

25 Sep 2025

Dear Dr Reddien,

Thank you for your patience while we considered your revised manuscript "The planarian dorsal-ventral boundary regulates anterior-posterior axis growth and patterning" for publication as a Short Report at PLOS Biology. Please accept my sincere apologies for the delays that you have experienced during this round of the peer review process. This revised version of your manuscript has been evaluated by the PLOS Biology editors, the Academic Editor and two of the original reviewers.

Based on the reviews, I am pleased to say that we are likely to accept this manuscript for publication, provided you satisfactorily address the remaining points raised by the reviewers. Please also make sure to address the following data and other policy-related requests that I have provided below (A-H):

(A) Your manuscript is being considered as a Short Report article type, which has a maximum of 4 main figures (https://journals.plos.org/plosbiology/s/what-we-publish#loc-short-reports). At this stage, we ask that you please reduce the number of main figures down to 4, either by combining main figures or moving a couple of figures to the Supplementary Information. I appreciate this may represent a significant amount of re-formatting so I have extended the revision timeframe by a week. Please do feel free to contact me if you would like to further discuss how best to reformat the figures.

(B) You may be aware of the PLOS Data Policy, which requires that all data be made available without restriction: http://journals.plos.org/plosbiology/s/data-availability. For more information, please also see this editorial: http://dx.doi.org/10.1371/journal.pbio.1001797

-Supplementary files (e.g., excel). Please ensure that all data files are uploaded as 'Supporting Information' and are invariably referred to (in the manuscript, figure legends, and the Description field when uploading your files) using the following format verbatim: S1 Data, S2 Data, etc. Multiple panels of a single or even several figures can be included as multiple sheets in one excel file that is saved using exactly the following convention: S1_Data.xlsx (using an underscore).

-Deposition in a publicly available repository. Please also provide the accession code or a reviewer link so that we may view your data before publication.

Figure 1B, 1D, S1H, S1J-K, S1L, S2A-B

(C) Thank you for providing the scRNA-seq-data in the SRA database (PRJNA1299474). However, I could not find the data when searching for it, so I would be grateful if you could check whether the accession number is correct or ensure that the data is publicly released.

(D) Please note that we cannot accept deposition of data in a Google drive. If you would like to provide data in a separate deposition, we ask that you use a public data repository (https://journals.plos.org/plosbiology/s/recommended-repositories).

(E) Please also ensure that each of the relevant figure legends in your manuscript include information on *WHERE THE UNDERLYING DATA CAN BE FOUND*, and ensure your supplemental data file/s has a legend.

(F) Please ensure that your Data Statement in the submission system accurately describes where your data can be found and is in final format, as it will be published as written there.

(G) Per journal policy, if you have generated any custom code during the course of this investigation, please make it available without restrictions. Please ensure that the code is sufficiently well documented and reusable, and that your Data Statement in the Editorial Manager submission system accurately describes where your code can be found.

(H) Please ensure that you are using best practice for statistical reporting and data presentation. These are our guidelines https://journals.plos.org/plosbiology/s/best-practices-in-research-reporting#loc-statistical-reporting and a useful resource on data presentation https://journals.plos.org/plosbiology/article?id=10.1371/journal.pbio.1002128

- If you are reporting experiments where n ≤ 5, please plot each individual data point.

We expect to receive your revised manuscript within three weeks.

*Published Peer Review History*

*Press*

Best regards,

Richard

Richard Hodge, PhD

rhodge@plos.org

Reviewer remarks:

Reviewer #1: In the revised version of the manuscript, now entitled "The planarian dorsal-ventral boundary regulates anterior-posterior axis growth and patterning", Maybrun et al. have addressed most of the main concerns. However, the SC-seq data of the DVB remain disconnected from the rest of the experiments. These data could potentially serve as the basis for functional studies in future work.

Before publication, the authors should carefully review the numbering of the figures cited in the text, as some of them are incorrect.

Reviewer #2: Many thanks the authors, the manuscript has been improved, and my difficulties have been largely dealt with. I think this should be published, although I have a couple minor points I'd like the authors to consider.

1) There are now two places in the manuscript where the authors distinguish between scattered "sporadic" expression of anterior makers and true anterior foci. The first is in their added language about the results of bcat-RNAi. The second is in their 9/22 outcomes with less focused anterior maker expression after cutting, bcat-RNAi and PD treatment to block DVB regeneration. In the latter, the authors give a fuller possible explanation, low level anterior induction that does not rise to the level or organizing foci. It might be helpful to have some of that language in the first discussion of the simple bcat-RNAi result, and try and define their criteria a bit.

2) A couple remaining clarifications about numbers:

Does the 18/21 in Figure S4D mean that in 3/21 cases the DVB still formed after cutting and PD treatment?

In the Fig. 5B/S4F data, if 12/22 don't have and 9/22 do have anterior gene expression after cutting, bcat-RNAi and PD treatment, what did the remaining 1/21 do?

---

## [Editor Report · Decision Letter 3]

22 Oct 2025

Dear Dr Reddien,

On behalf of my colleagues and the Academic Editor, Konrad Basler, I am pleased to say that we can accept your manuscript for publication, provided you address any remaining formatting and reporting issues. These will be detailed in an email you should receive within 2-3 business days from our colleagues in the journal operations team; no action is required from you until then. Please note that we will not be able to formally accept your manuscript and schedule it for publication until you have completed any requested changes.

PRESS

Best wishes, 

Richard

Richard Hodge, PhD

rhodge@plos.org

PLOS
